# Glutamine metabolism inhibition has dual immunomodulatory and antibacterial activities against *Mycobacterium tuberculosis*

Sadiya Parveen[1], Jessica Shen[1], Shichun Lun[1], Liang Zhao [ ][2], Jesse Alt[3], Benjamin Koleske[1], Robert D. Leone[2,4], Rana Rais[5,6,7], Jonathan D. Powell[4,10], John R. Murphy[1], Barbara S. Slusher[2,3,4,5,6,7,8,9] & William R. Bishai [ ][1] ✉

As one of the most successful human pathogens, *Mycobacterium tuberculosis* (*Mtb*) has evolved a diverse array of determinants to subvert host immunity and alter host metabolic patterns. However, the mechanisms of pathogen interference with host metabolism remain poorly understood. Here we show that a glutamine metabolism antagonist, JHU083, inhibits *Mtb* proliferation in vitro and in vivo. JHU083-treated mice exhibit weight gain, improved survival, a 2.5 log lower lung bacillary burden at 35 days post-infection, and reduced lung pathology. JHU083 treatment also initiates earlier T-cell recruitment, increased proinflammatory myeloid cell infiltration, and a reduced frequency of immunosuppressive myeloid cells when compared to uninfected and rifampin-treated controls. Metabolomic analysis of lungs from JHU083-treated *Mtb*-infected mice reveals citrulline accumulation, suggesting elevated nitric oxide (NO) synthesis, and lowered levels of quinolinic acid which is derived from the immunosuppressive metabolite kynurenine. JHU083-treated macrophages also produce more NO potentiating their antibacterial activity. When tested in an immunocompromised mouse model of *Mtb* infection, JHU083 loses its therapeutic efficacy suggesting the drug's host-directed effects are likely to be predominant. Collectively, these data reveal that JHU083-mediated glutamine metabolism inhibition results in dual antibacterial and host-directed activity against tuberculosis.

Tuberculosis (TB) remains a leading infectious disease killer causing ~1.3 million deaths worldwide in 2022 alone[1]. The causative bacterium, *Mycobacterium tuberculosis* (*Mtb*), is one of the most successful global pathogens, and it has evolved multiple mechanisms to evade and suppress the host immune system[2]. While effector CD4+ T-cells are considered one of the most potent factors in host containment of the pathogen, individuals with active TB disease fail to mount adequate cellular immune responses[3,4], and the mechanisms by which *Mtb*

infection leads to failed effector T-cell responses remain poorly understood.

Suppression of effector T-cell immunity is also one of the hallmarks of the tumor microenvironment[5]. Recently, a glutamine (Gln) metabolism antagonist drug, JHU083, has been shown to reprogram host immunometabolic signatures and improve effector T-cell immunity in several murine tumor models[6-8]. High Gln metabolism in tumors were found to be closely associated with an abundance of

immunosuppressive cells such as myeloid-derived suppressor cells (MDSCs) and regulatory T-cells (Tregs) with a concomitant inhibition of effector T cells[9,10]. JHU083 administration not only reduced Gln utilization but also enhanced the antitumor activity of effector T cells[6,7]. JHU083 and a closely related drug, Sirpiglenastat, which is in human clinical trials, are prodrugs of the glutamine antagonist, 6-diazo-5-oxo-L-norleucine (DON), which has shown to have anticancer activity in preclinical models and early clinical trials[8,11]. DON irreversibly inhibits glutamine-utilizing enzymes including glutaminases, glutamine synthetases as well as multiple glutamine amidotransferases involved in the biosynthesis of purines, pyrimidine, coenzymes, hexosamines and amino acids[12]. While DON showed robust antitumor activity in both preclinical and early clinical trials, its development was hampered by dose-limiting toxicities which were mainly gastrointestinal (GI) tract-related[12]. To circumvent these GI toxicities and enhance its therapeutic index, prodrugs of DON were designed to preferentially deliver DON to tumors while minimizing its release in the gut[11,13]. One such prodrug, JHU083, is orally available and is converted to DON by ubiquitous peptidases and esterases, has fewer side effects, and has shown the ability to enhance T-cell immunity in the tumor microenvironment[8,11]. Since blunted effector T-cell immunity are characteristics of both the cancer microenvironment and the tuberculous granuloma[14], we hypothesized that JHU083 might also enhance effector T-cell immunity by a similar immunometabolic reprogramming mechanism and thereby improve host containment of TB disease progression.

In support of this hypothesis, early seminal work by Horwitz and colleagues indicated that inhibition of bacterial glutamine (Gln) metabolism may have direct antibacterial activity[15–17]. This group showed that *Mtb* possesses four glutamine synthetase enzymes. Among these four proteins, only GlnA1 is secreted, and was found to be essential for *Mtb* survival in vitro and in vivo[17]. In addition, Horwitz

et al. also showed that methionine sulfoximine (MSO), an irreversible inhibitor of glutamine synthetase enzymes, has direct antimycobacterial activity with an *Mtb* MIC of 50 µM on solid medium[15]. MSO treatment also reduced the bacillary burden in the guinea pig TB model[15]. These observations, together with the aforementioned immunometabolic reprogramming by a glutamine metabolism antagonist, suggested to us that *Mtb* may employ its extracellular GlnA1 enzyme as a virulence factor to increase Gln metabolism in the granuloma microenvironment, thereby creating an immunosuppressive cellular milieu as has been shown in tumors. We, therefore, hypothesized that JHU083 may serve as a host-directed therapy against TB disease by reprogramming Gln metabolism in granulomas. In light of the known direct antibacterial effects of the glutamine synthetase inhibitor MSO, we considered the possibility that JHU083 might possess direct antibacterial activity as well. Here, we show that glutamine antagonism may be beneficial in host *Mtb* containment through a combination of immunomodulatory and antibacterial killing.

## Results

### Both DON and JHU083 exhibit modest antibacterial activity against *Mtb*

Based on the earlier work of Horwitz and colleagues showing the essentiality of GlnA1, the major *Mtb* glutamine synthetase, we hypothesized that JHU083, a glutamine synthetase inhibitor, may also inhibit *Mtb* growth[18]. Using the Alamar blue assay, we tested the antimycobacterial activity of JHU083, DON, and MSO, a known inhibitor of *Mtb* GlnA1[15,19] (Fig. 1a, b). Both JHU083 and DON inhibited *Mtb* H37Rv growth in vitro with an MICs of 1–2 µg/ml which was two to four times more potent than MSO (MIC = 4 µg/ml) (Fig. 1c). RIF was used as the positive control in these assays (MIC = 0.25 µg/ml) (Fig. 1c). Additionally, exogenous glutamine supplementation, even at 5-fold-molar

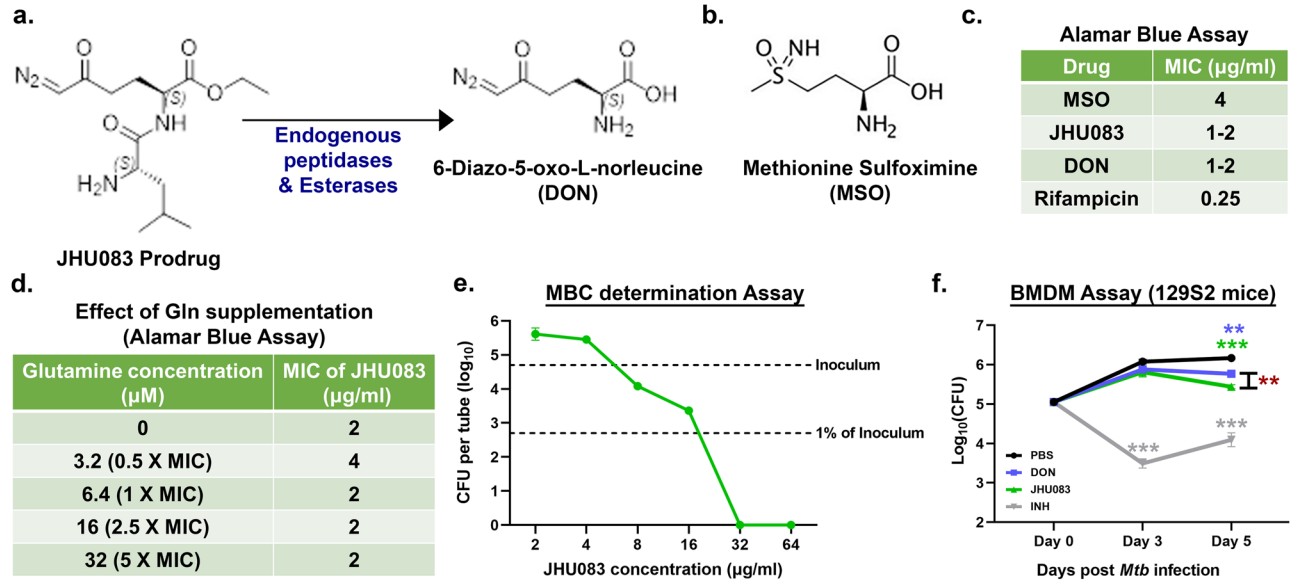

**Fig. 1 | JHU083 has direct antimycobacterial activity in vitro. a** Chemical structures of the prodrug JHU083 and the active drug DON. Endogenous host esterases and peptidases convert JHU083 into DON. **b** Chemical structure of MSO. **c** Table showing the minimum inhibitory concentrations (MIC) values of the drugs against the *Mtb* H37Rv strain determined using the Alamar blue assay. **d** Table depicting the effect of glutamine (Gln) supplementation on the antimycobacterial activity of JHU083 using the Alamar blue assay. For each assay, a fixed concentration of Gln was used as shown with the assumed MIC of JHU083 being 2.0 µg/ml or 6.4 µM. **e** Graph showing the results of the minimal bactericidal concentration (MBC) determination assay. The top dotted line represents the starting inoculum of the *Mtb* culture while the bottom dotted line represents 1% of the initial inoculum.

**f** Antibacterial activity of drugs against *Mtb* growing within bone-marrow derived macrophages (BMDMs). IFNγ-activated BMDM from 129S2 mice were infected with *Mtb* H37Rv at an MOI of 2. They were then treated with 10 µg/ml of either DON or JHU083 (5x the MIC of each drug assuming the MIC is 2.0 µg/ml). Isoniazid (INH) at 1.28 µg/ml was used as the positive control. The cells were lysed at indicated time points and plated on 7H11 selection plates. The assay was performed as described in the "Methods". Data were plotted as mean ± SEM. Statistical significance was calculated using a two-tailed student *t* test considering unequal distribution. The exact *p*-values are provided in the Source Data file. * < 0.05, ** < 0.01, *** < 0.001. All the experiments were performed in triplicate.

excess, did not reduce antibacterial activity of JHU083 (Fig. 1d). We also determined the minimum bactericidal concentration (MBC), colony forming units (CFU)-based killing assay, and found that JHU083 has an MBC value of 32 μg/ml (Fig. 1e). Thus, JHU083 is bacteriostatic at lower concentrations (MIC 1–2 μg/ml) and bactericidal at higher concentrations (MBC 32 μg/ml).

We also determined the antibacterial activity of DON and JHU083 ex vivo in *Mtb* H37Rv infected murine bone-marrow derived macrophages. DON and JHU083 reduced *Mtb* bacterial burden by 0.3 and 0.7 $\log_{10}$ units respectively after 5 days of treatment ($P < 0.05$; Fig. 1f). Both 1X and 10X MIC of the active drug resulted in a similar decline in bacillary burden (Fig S1a). Lack of dose-dependent effects is likely due to either the saturation of the transport machinery available for the drug entry in the macrophages or 1X MIC caused the maximum inhibition possible. The standard TB drug isoniazid reduced the bacterial burden by 1.2 $\log_{10}$ units ($P < 0.001$; Fig. 1f). To confirm that the decreased bacterial burden in macrophages was due to drug antimycobacterial activity and not cytotoxic activity of the drug, we evaluated the viability of macrophages in the presence of these same drugs and found no decrease in macrophage viability by MTS assay (Fig S1b). These results show that both DON and JHU083 possess direct antibacterial activity against *Mtb*.

### JHU083 reduced disease burden in murine TB models

We next evaluated the therapeutic efficacy of JHU083 in *Mtb*-infected 129S2 and C3HeB/FeJ mice. These strains, when infected with *Mtb*, accumulate a higher frequency of myeloid derived suppressor cells (MDSCs) compared to relatively less susceptible strains such as Balb/c[7,20,21]. MDSCs are also one of the major targets of JHU083 action[7]. We challenged 129S2 mice with ~275 CFU of *Mtb* H37Rv via the aerosol route, and randomized mice into three groups: (1) untreated, (2) JHU083 treatment (1 mg/kg daily), and (3) RIF treatment (12.5 mg/kg daily) (Fig. 2a) with treatment initiation one day after *Mtb* challenge. Lungs were harvested from all three groups at weeks 2 and 5, homogenized, and plated on 7H11 agar plates to quantify the bacillary burden. While JHU083 did not reduce lung CFU at week 2, it reduced lung bacterial burden by 1.9 $\log_{10}$ units after 5 weeks of treatment compared to untreated mice ($P < 0.05$; Fig. 2b). Based on previous preclinical studies with JHU083, two different dosing regimens; (1) Daily (1 mg/kg dose per day for first week, followed by 0.3 mg/kg daily) and (2) Alternate (1 mg/kg dose per day for first week, followed by 1 mg/kg on Mon, Wed and Fri) were tested[22–25]. Both testing regimens resulted in a similar reduction in the lung bacillary burden (Fig S2a) and lung weight (Fig S2b). We measured the level of DON in *Mtb*-infected lungs and found it to be 0.856 nMoles/g lung tissue consistent with earlier reports[22–25]; thus the measured lung concentration of DON is >10,000-fold below the MIC value of either DON (MIC 5.8–11.6 μM) or JHU083 (MIC 3.2–6.4 μM) (Fig. 2c). In addition to 129S2 mice, we also measured JHU083 activity in *Mtb*-infected C3H and C3HeB/FeJ mice, the latter strain being well-known to develop necrotic granulomas closely resembling the granulomas observed in human patients[26–28]. In C3HeB/FeJ mice JHU083 treatment reduced lung bacterial burden by 1.0 $\log_{10}$ unit compared to untreated mice ($P < 0.05$; Fig S3a). In contrast, JHU083 treatment did not have therapeutic efficacy in C3H mice (Fig S3b).

Consistent with lowered bacterial burden, JHU083 treatment significantly prolonged 129S2 mouse survival with a median time to death (MTD) of greater than 60 days compared with untreated controls which showed an MTD of 35.5 days ($P < 0.01$; Fig. 2d). JHU083-treated mice gained weight to a greater degree than untreated controls during treatment ($P < 0.0001$; Fig S4a) and had consistently lower lung weights than the untreated mice ($P < 0.001$; Fig. 2e). Quantitative histopathological examination of lungs confirmed significant reduction in the granulomatous lesions in JHU083-treated group compared to the untreated mice ($P < 0.05$; Fig. 2g). While there was ~50% reduction in granulomatous lesions in both C3HeB/FeJ and 129S2 mice strains, the

change reached statistical significance only in case of C3HeB/FeJ mice (Figs. 2h, i and S5). The positive control group, RIF-treated mice, exhibited lowered lung bacillary burden, lung weight and lesions concomitant with increased body weight and survival compared to the untreated mice (Figs. 2b, e, h, S4b and S4c). 129S2 mice were chosen for the rest of the experiments owing to a more pronounced reduction in the bacillary burden and lowered intragroup variability. These observations demonstrate that JHU083 inhibits *Mtb* growth in lungs, prolongs survival, and reduces lung pathology in at least two murine TB models.

### JHU083 shows no therapeutic efficacy against *Mtb* in immunocompromised mice

While the Alamar blue assay suggested that JHU083 has direct antibacterial activity against *Mtb*, murine cancer models have shown that JHU083 also exerts beneficial immunomodulatory effects[6,7]. To delineate a more precise mechanism of action, we evaluated the therapeutic efficacy of JHU083 in immunocompromised mice, *Mtb*-infected Balb/c SCID mice. We infected SCID mice with ~40 CFU of *Mtb* H37Rv on day 0, and individual groups were treated with PBS, JHU083 (1 mg/kg daily), or RIF (12.5 mg/kg daily) starting on day 1(Fig. 3a). After 5 weeks of infection and treatment, we observed no reduction in the lung bacillary burden in the JHU083-treated mice compared to untreated controls while RIF treatment reduced the lung *Mtb* CFU burden by 4.24 $\log_{10}$ units (Fig. 3b). Additionally, both untreated and JHU083-treated SCID mice showed significant weight loss, while RIF-treated mice continued to gain weight ($P < 0.01$, Fig. 3c). Importantly, JHU083-treated SCID mice died at the same median time to death as untreated mice (MTD 41days), while RIF significantly prolonged survival with an MTD of 73 days ($P < 0.0001$; Fig. 3d). These observations strongly suggest that an intact immune system is required for the therapeutic efficacy of JHU083 against *Mtb*. Despite the drug's direct antibacterial activity, these data show that JHU083 is likely to exert most of its therapeutic benefit via immunomodulation rather than direct bacterial killing.

### Glutamine metabolism inhibition leads to early recruitment of activated T-cells to the lungs during *Mtb* infection

Several studies in the cancer field have reported that the immunomodulatory effects of glutamine metabolism inhibitors (JHU083 included) are driven by their impact on T-cells[6,7,29]. Specifically, Leone et al. showed that JHU083 metabolically reprograms T-cells and increases the frequency of long-lived, activated effector T-cells in murine cancer models[6]. We, therefore, asked if JHU083 would improve effector T-cell immune responses in *Mtb*-infected 129S2 mice. To examine this, we harvested lungs from untreated, JHU083-treated, and RIF-treated mice groups and performed multicolor flow cytometry as described in Methods. Various T and B-cell subsets were identified, as shown in Fig S6, S7, and S8. At week 2 post-infection and treatment, the lungs from JHU083-treated mice showed a statistically significant 25% higher frequency of CD4+ T-cells (mean = 36.9% of live CD45+) compared to both untreated (mean = 29.5% of live CD45+) and RIF-treated mice (mean = 24.5% of live CD45+) (Fig. 4a). These CD4+ T-cells also showed significant enrichment for the cytokines released by activated T-cells, such as tumor necrosis factor (TNFα) and interferon-gamma (IFNγ) (Fig. 4b, c) and a significant decrease in expression of the immunosuppressive marker interleukin-10 (IL-10; Fig. 4d). We also found that CD4+ T-cell expansion in JHU083-treated mice was driven by significant increases in the frequencies of naïve (CD4+ CD62L+ CD44-) and follicular helper T-cells (CD4+ BCL6+) compared to untreated and RIF-treated controls (Fig. 4e, f). Accordingly, both CD4+ and CD8+ T-cells exhibited an increase in the expression of CD62L and BCL6 in JHU083-treated mice (Figs. 4g, h and S9a–S9d). However, there was no overall increase in the frequency of CD8+ T-cells (Fig. S9e). Consistent with naïve T-cell phenotype, both CD4+ and CD8+ T-cells

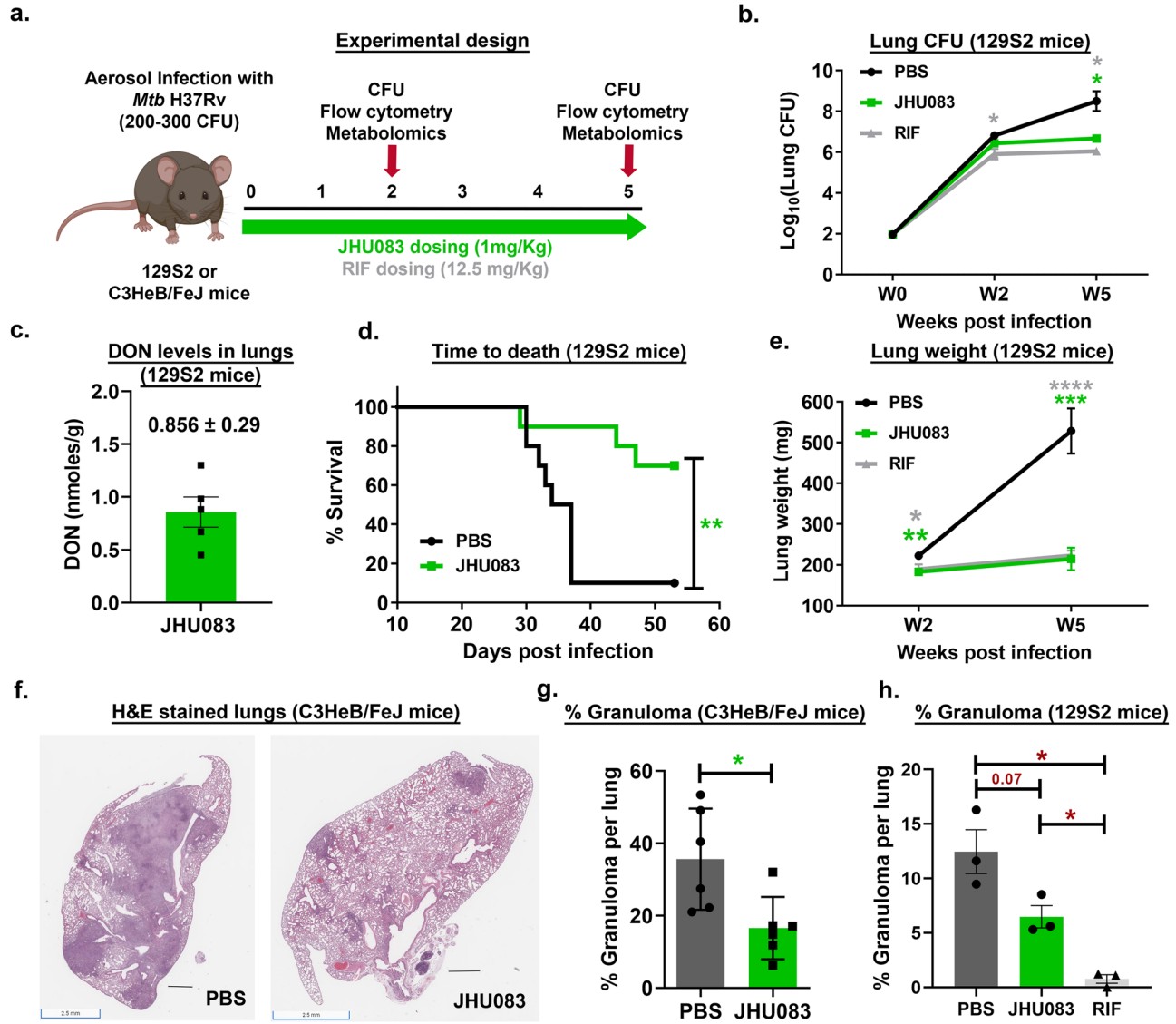

**Fig. 2 | JHU083 administration reduces *Mtb* proliferation and lung pathology in mice. a** Schematic of the mouse experiments. Six to ten weeks old 129S2 or C3HeB/FeJ mice (*n* = 4–10/group) were aerosol infected with ~200–300 CFU of *Mtb* H37Rv. The mice were treated with JHU083 or RIF by oral gavage starting one day after infection. 1 mg/kg of JHU083 was given daily for the first five days, and then the dose was reduced to 0.3 mg/kg daily (5/7, M-F). 12.5 mg/kg RIF was given daily orally for 5-weeks. **b** Lung bacillary burden in 129S2 mice treated with JHU083 or PBS or RIF (*n* = 5/group for week 2 & *n* = 4/group for week 5). Mice were sacrificed at day 0, week 2, and week 5 post-infection/treatment. The lungs were harvested, homogenized, diluted, and plated on 7H11 selection plates. After 21–25 days, the colonies were counted, and counts were transformed into log_{10} values and plotted. **c** DON levels in lungs of JHU083-treated, *Mtb*-infected 129S2 mice (*n* = 5/group) at 2 weeks post-infection, 30–40 min after receiving a 1 mg/kg dose of JHU083 (the tissue Tmax of JHU083 is 30 min post-oral dosing; full dosing details are in the Methods). DON levels were determined using LC/MS-based targeted metabolomics. **d** Survival of 129S2 mice treated with JHU083 or PBS. **e** Gross lung weight of 129S2 mice

(*n* = 10/group) at weeks 2 and 5 post infection/treatment. **f** Histopathology of lungs isolated from C3HeB/FeJ mice infected with *Mtb* H37Rv at week 4.5 post-infection/treatment. The lungs were formalin fixed, sectioned, and stained with hematoxylin and eosin (H&E). **g** Quantitation of the lung granuloma areas in C3HeB/FeJ mice infected with *Mtb* H37Rv at week 4.5 post-infection/treatment (*n* = 6).
**h** Quantitation of the lung granuloma areas in the 129S2 mice lungs infected with *Mtb* H37Rv at week 5 post-infection/treatment (*n* = 3). Both total granuloma area (GA) and lung area (LA) areas were measured using ImageScope software (Leica). The percent granuloma area (%GA) was calculated using the formula (%GA = (GA X 100)/LA). Data were plotted as mean ± SEM. Statistical significance was calculated using a two-tailed student *t*-test considering unequal distribution. For survival curve, log-rank (Mantel-Cox) and Gehan-Breslow-Wilcoxon tests were used and yielded similar *p*-value. The exact *p*-values are provided in the Source Data file. * < 0.05, ** < 0.01, *** < 0.001, **** < 0.0001. CFU stands for colony-forming units. All the experiments were repeated at least twice. The mouse clipart shown in 2a was created using Biorender.com (Tornonto, Canada) by SP.

had lower expression of Klrg1, a terminal differentiation marker, in the JHU083-treated group (Fig. 3i, j). In keeping with an increase in follicular helper T cells, we also found higher frequencies of mature and memory B-cells in both JHU083- and RIF-treated animals compared to untreated controls (Fig. 3k, l). Interestingly, most of the differences in T-cell immune responses in the JHU083-treated group subsided by week 5 (Fig. S10a–S10j). In uninfected mice, JHU083 treatment for two weeks lowered the frequency of B-cells (Fig. S11a) but had no effect on

any of the T-cell subsets tested (Fig. S11b–S11k). Overall, the data indicates that JHU083 has an early but transient effect on T-cell recruitment to *Mtb*-infected lungs.

## Glutamine metabolism inhibition modulates macrophages towards an inflammatory signature
Myeloid cells, including macrophages and MDSCs, are known to play an important role in *Mtb* containment[14]. Since JHU083 is also known to

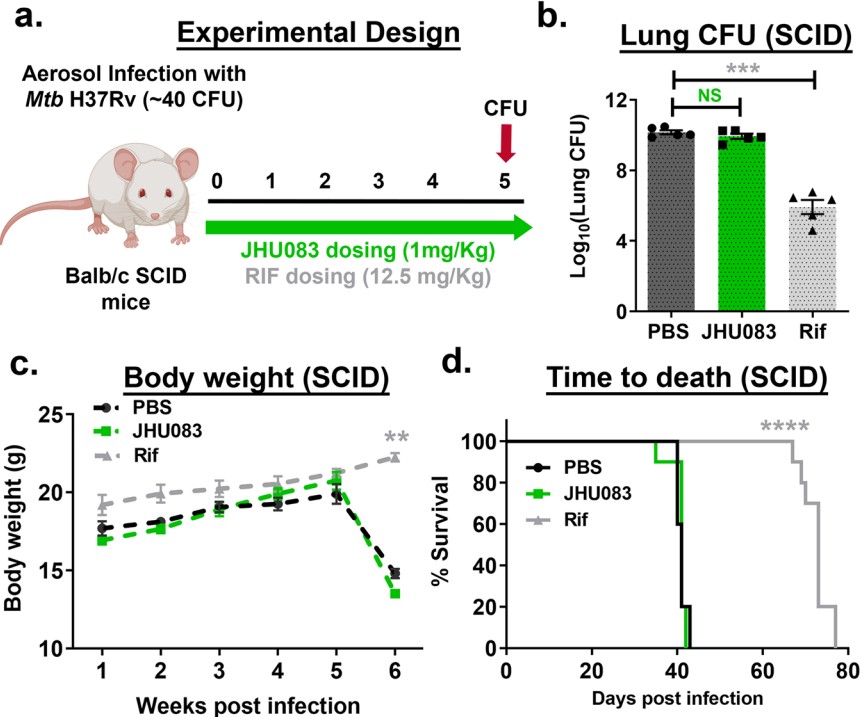

**Fig. 3 | JHU083 lacks therapeutic efficacy in *Mtb*-infected SCID mice.**
**a** Schematic of the SCID mouse experiments. Six to ten weeks old Balb/c SCID mice
($n = 5–10$/group) were aerosol infected with ~40 CFU of *Mtb H37Rv*. The mice were
then treated with JHU083 or rifampin (RIF) by oral gavage starting one day after
infection. 1 mg/kg JHU083 was given daily for the first 5 days and then the dose was
reduced to 0.3 mg/kg daily (5/7, M-F). **b** Lung CFU burden in SCID mice ($n = 5$/
group) treated with JHU083, RIF, or PBS. Mice were sacrificed at week 5 post-
infection/treatment. The lungs were harvested, homogenized, diluted, and plated
on 7H11 selection plates. After 21–25 days, the colonies were counted, and counts
were transformed into $\log_{10}$ values and plotted. **c** Body weights of SCID mice

treated with JHU083, RIF, or PBS ($n = 10$/group). **d** Survival of SCID mice treated
with JHU083, RIF, or PBS ($n = 10$/group). Data were plotted as mean ± SEM. Statis-
tical significance was calculated using a two-tailed student $t$ test considering
unequal distribution. For survival curve, log-rank (Mantel-Cox) and Gehan-Breslow-
Wilcoxon tests were used and yielded similar $p$-value. The exact $p$-values are pro-
vided in the Source Data file. * < 0.05, ** < 0.01, *** < 0.001, **** < 0.0001. CFU stands
for colony-forming units. NS stands for non-significant change, $p$-value was >0.05.
The experiment was repeated twice. The mouse clipart shown in 3a was created
using Biorender.com (Tornonto, Canada) by SP.

reprogram myeloid cells and reinforce their host-protective functions
in murine cancer models, we investigated the impact of JHU083
administration upon lung myeloid cell populations. Using the same
model in 129S2 mice (Fig. 2a), we performed flow cytometry on lung
cells from mice sacrificed at weeks 2 and 5 post-infection/treatment as
described in Methods. The gating strategy is described in Fig S12
and S13.

We found that total lung CD11b+ myeloid cells were significantly
increased at week 5, but not at week 2 in both JHU083- and RIF-
treated mice compared to untreated mice (Fig. 5a). We then eval-
uated three major myeloid cell subsets: (1) alveolar macrophages
(AM; CD45+ SiglecF+); (2) Interstitial macrophages (IM; CD45+ CD11b+
F4/80+), and (3) myeloid-derived suppressor cells (MDSCs; CD45+
CD11b+ Ly6G/Ly6C). We detected a two-fold decrease in the fre-
quency of AM in JHU083-treated mice compared to untreated mice at
week 2, but not at week 5 (Fig. 5b, $P = 0.05$). At the 2-week timepoint,
the JHU083-treated group exhibited higher expression of the co-
stimulatory molecule CD86 (Fig. 5c) and lower expression of inhibi-
tory receptor CD206 than untreated controls (Fig. 5d), consistent
with the AMs present in the lung at 2 weeks being enriched for a
proinflammatory phenotype.

In contrast to AMs, the IM cell frequencies were unchanged in
JHU083-treated animals at 2 weeks compared to uninfected animals,
but significantly increased (2-fold) in JHU083-treated animals at the
late timepoint of week 5 (Figs. 5e, S14a). In spite of this late recruitment
of IMs to the lungs in JHU083-treated mice, the expression of either
CD86 or CD206 remained constant, suggesting a balance of M1- and
M2-like IMs entering the lungs (Figs. S14b and S14c).

We then analyzed two prominent subsets of MDSCs in these
murine lung samples: (1) Monocytic MDSCs (mMDSCs; CD11b+ Ly6G-
Ly6C$^{High}$) and (2) Granulocytic MDSCs (gMDSCs; CD11b+ Ly6G+
Ly6C$^{Low}$). mMDSCs were present at significantly higher frequencies at
both weeks 2 and 5 compared to the untreated group (Fig. S14d), while
gMDSCs did not significantly differ in the JHU083-treated animal
compared with the other groups (Fig. S14e). However, we found that
both IL-10-expressing mMDSCs and gMDSCs were present in reduced
frequencies at the early 2-week timepoint, but subsequently were not
significantly different from the other groups at 5 weeks (Figs. 5f, g, S14f
and S14g). Thus, our analysis of lung myeloid cells revealed an early
reduction of *Mtb*-permissive AMs which later normalized, a late influx
(5 weeks) of *Mtb*-restrictive IMs, and an early reduction of immuno-
suppressive subtypes, namely IL10+ mMDSCs, and IL-10+ gMDSCs.
Interestingly, even in the absence of *Mtb* challenge JHU083 treatment
still led to the enrichment of IMs and mMDSCs without altering the
frequencies of AM and gMDSCs (Fig. S15).

These myeloid cell shifts mediated by JHU083-treatment suggest
that the drug promotes enrichment of cells restrictive of *Mtb* growth
and reduces those associated with immunosuppression.

## JHU083 drives host-protective metabolic changes in *Mtb*-infected lungs
Since JHU083 has been shown to reprogram metabolic pathways of
both cancer and immune cells by blocking glutamine metabolism[6,7], we
hypothesized that JHU083 administration would alter metabolic
pathways in *Mtb*-infected lungs. To investigate this, we performed LC/
MS-based metabolomics of total lung tissues from untreated, JHU083-

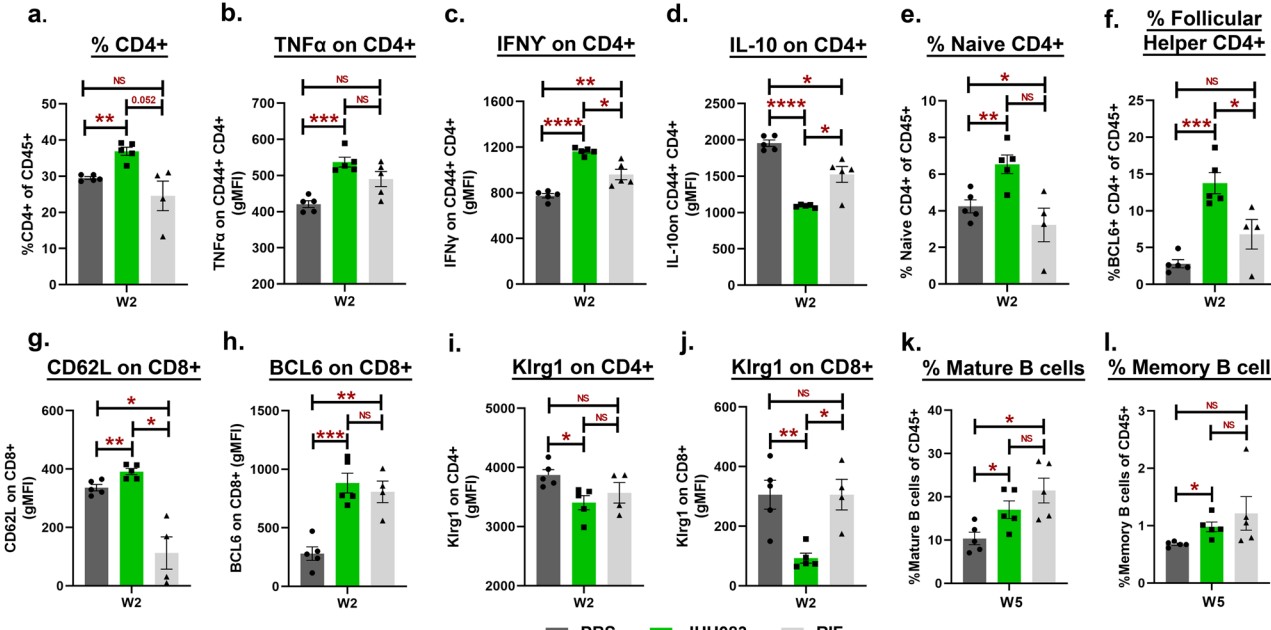

**Fig. 4 | JHU083 treatment elicits early recruitment of T-cells in the lungs.** As described in Fig. 2a, *Mtb*-infected female 129S2 mice (*n* = 5/group) were treated with JHU083, RIF, and PBS daily starting on day 1 post infection. Mice were sacrificed on week 2 and week 5, and the lungs were harvested. Single cell suspensions of the lungs from the three treatment groups were stained with appropriate antibodies and analyzed using multicolor-flow cytometry. We found differences in the frequencies or geometric mean fluorescence intensities (gMFI) for: (**a**) CD4⁺ T-cells, (**b**) TNFα expression on activated (CD44⁺) CD4⁺ T-cells, (**c**) IFNγ expression on activated (CD44⁺) CD4⁺ T-cells, (**d**) IL-10 expression on activated (CD44⁺) CD4⁺ T-cells, (**e**) naïve CD4⁺ T-cells, (**f**) follicular helper (BCL6⁺) CD4⁺ T-cells, (**g**) CD62L expression on CD8⁺ T-cells, (**h**) BCL6 expression on CD8⁺ T-cells, (**i**) Klrg1 expression on CD4⁺ T-cells, (**j**) Klrg1 expression on CD8⁺ T-cells, (**k**) mature B-cells (CD19⁺ CD27⁻ CD138⁻), and (**l**) memory B-cells (CD19⁺ CD27⁺ CD138⁻). The X-axis indicates the lung harvest timepoint. All T-cell data (**a–j**) came from the lungs harvested at week 2 (W2) post-infection/treatment while B-cell data (**k–l**) was generated from lungs harvested at week 5 (W5) post-infection/treatment. For all graph panels, *n* = 5/group except RIF-treated group in graph (**e–j**) due to loss of a sample during processing (*n* = 4). Data were plotted as mean ± SEM and are shown as the frequency of CD45⁺ population. gMFI was mostly used for low abundance cell surface markers and transcription factors. Statistical significance was calculated using a two-tailed student *t*-test considering unequal distribution. The exact *p*-values are provided in the Source Data file. * < 0.05, ** < 0.01, *** < 0.001, **** < 0.0001. NS stands for non-significant change, *p*-value was >0.05. The experiment was repeated two times.

and RIF-treated animals at weeks 2 and 5. We detected a total of 144 metabolites and found alterations in the levels of >100 metabolites among the groups pointing toward a complex metabolic reprogramming (Supplementary Data 1). JHU083-treated mice exhibited maximal changes in metabolite levels at week 2 post-infection and treatment (Fig. S16) in contrast to the RIF-treated group that showed minimal changes at week 2 (Fig. S17). One of the most notable changes in JHU083-treated lungs was alterations in the arginine metabolism with accumulation of citrulline (1.4-fold increase; *P* = 0.04; Fig. 6a) accompanied by significantly lower levels of other arginine metabolites including ornithine, polyamines, S-adenosyl-homocysteine, creatine, and agmatine (Figs. 6e, S18a). To confirm whether higher citrulline level leads to higher NO production, a Griess-reagent based colorimetric NO assay was performed with BMDMs. JHU083-treated macrophages produced more NO compared to both untreated (*P* = 0.026; PBS = 181 μM vs JHU083 = 330 μM) and INH-treated (*P* = 0.21; PBS = 216 μM vs JHU083 = 330 μM) groups (Fig. 6b). There was no difference in the level of NO production between PBS- and INH-treated groups (*P* = 0.68; PBS = 181 μM vs JHU083 = 216 μM; Fig. 6b). This data clearly demonstrated that JHU083 treatment leads to elevated levels of NO in macrophages potentially via iNOS.

We also detected alterations in the metabolism of tryptophan with accumulation of 5-hydroxy-3-indoleacetic acid (5-HIAA; 2.9-fold increase, *P* = 0.008; Fig. 6c) and ~60% reduction in the level of quinolinic acid which is a byproduct of the immunosuppressive metabolite kynurenine (*P* = 0.02; Figs. 6d, f, S18b). However, kynurenine could not be detected in the samples. As a proxy of Trp-Kyn pathway, the level of the first enzyme in the pathway, IDO1, was quantified in the whole lung lysate using western blotting. The IDO1 levels were found to be similar

among all three treatment groups too (Fig. S19). Surprisingly, we did not detect statistically significant changes in either Gln or Gln:Glu ratio in whole lungs between untreated and JHU083-treated groups (Fig. S20). Overall, the data shows that JHU083 treatment altered the metabolism of two immunologically important amino acids, arginine, and tryptophan, in directions that appear to drive enhanced anti-*Mtb* host immune responses (Fig. 7).

## Discussion

*Mtb*, a sophisticated intracellular pathogen, deploys numerous strategies to overcome the host immune system, starting from interfering with immune cell recruitment, inhibiting their host-protective functions and compromising their metabolic fitness. Interestingly, little is known about the effect of *Mtb* infection upon host metabolic pathways or how these alterations contribute to TB pathogenesis. Here, we show that JHU083-mediated glutamine metabolism inhibition prevents *Mtb* proliferation both in vitro and in vivo. JHU083-treated *Mtb*-infected mice had lowered lung bacillary burden, gained weight, and lived longer with improved lung histopathology. JHU083 treatment also caused an earlier onset of T-cell recruitment and a reduced frequency of immunosuppressive myeloid cells in the lungs. Metabolomics analysis revealed that JHU083-treated lungs accumulated citrulline (suggesting greater host-protective NOS activity) and lowered quinolinic acid (a byproduct of the immunosuppressive molecule kynurenine). Additionally, JHU083-treated *Mtb*-infected macrophages also released more NO compared to untreated controls. Overall, JHU083-treated animals exhibited an improved anti-*Mtb* immune response.

JHU083 is a prodrug that is converted to 6-diazo-5-oxo-norleucine (DON) in the serum and the tissues in the presence of esterases and

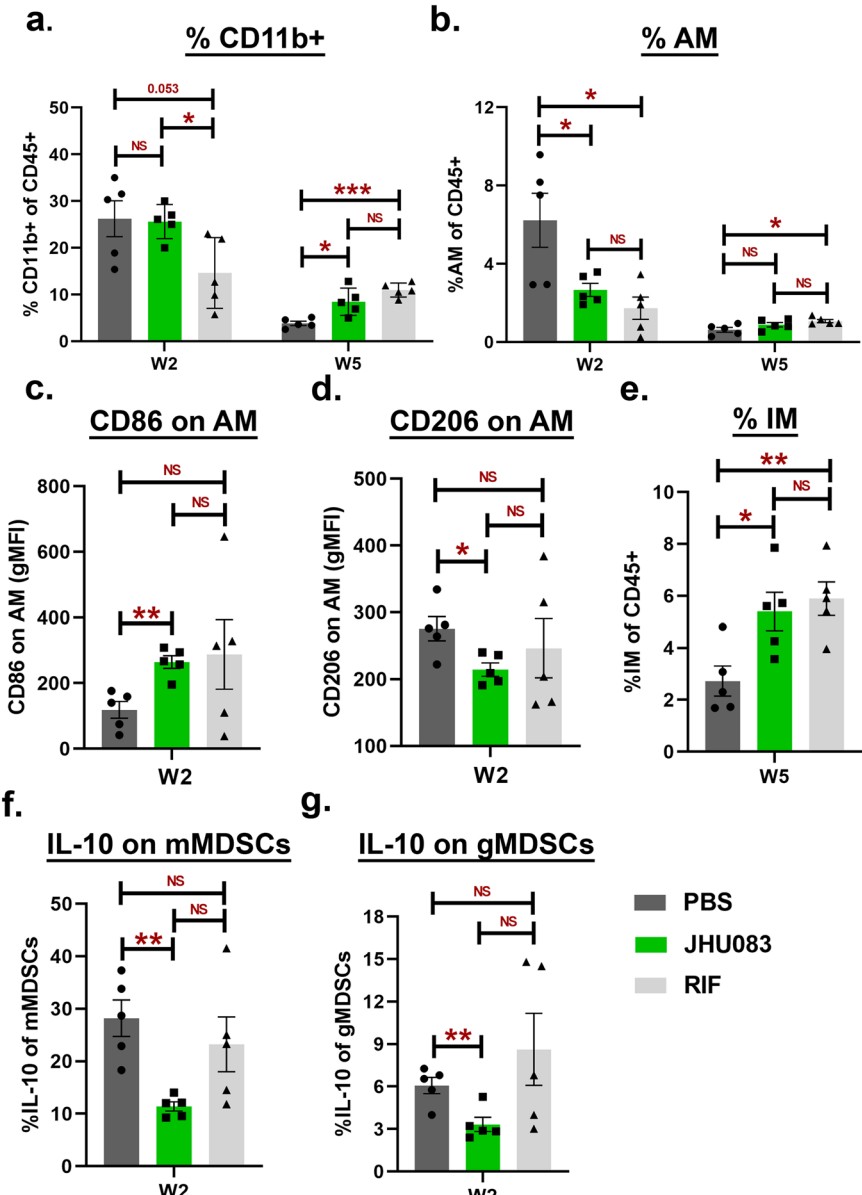

**Fig. 5 | JHU083 treatment modulates lung myeloid cell populations.** As described in Fig. 2a, *Mtb*-infected 129S2 mice were treated with JHU083, RIF, or PBS every day starting at day 1 post-infection (*n* = 5/group). The mice were sacrificed on week 2 and week 5, and the lungs were harvested. Single cell suspension of the lungs from all three treatment groups were stained with appropriate antibodies and analyzed using multicolor-flow cytometry (*n* = 5 mice). Details are provided in "Methods" section. We found differences in the frequencies or gMFI values for (**a**) CD11b+ myeloid cells, (**b**) alveolar macrophages (AM, CD11b+ SiglecF+), (**c**) CD86 expression upon AM, (**d**) CD206 expression upon AM, (**e**) interstitial macrophages (IM, CD11b+ SiglecF- F4/80+), (**f**) IL-10 expression upon monocytic myeloid-derived suppressor cells (mMDSCs, CD11b+ Ly6G- Ly6CHigh), (**g**) IL-10 expression upon granulocytic myeloid-derived suppressor cells (gMDSCs, CD11b+ Ly6G+ Ly6Clow). The X-axis indicates the lung harvest timepoint. Data were plotted as mean ± SEM and are shown as the frequency of CD45+ population. gMFI was mostly used for low abundance cell surface markers and transcription factors. Statistical significance was calculated using a two-tailed student *t*-test considering unequal distribution. The exact *p*-values are provided in the Source Data file. * < 0.05, ** < 0.01. NS stands for non-significant change, *p*-value was >0.05. The experiment was repeated twice.

peptidases. DON exhibits high structural similarity to glutamine, blocks several reactions that either generate or require glutamine and has been extensively studied as a potential cancer therapeutic[12,30]. However, initial clinical trials were hampered by the dose-limiting gastrointestinal (GI) toxicity of DON[12]. Extensive pharmacokinetic studies in multiple preclinical models have demonstrated that the prodrug strategy not only reduces GI toxicity but also increases its bioavailability[8,11,13,31,32]. In the tumor microenvironment, elevated glutamine metabolism has been associated with increased immunosuppression leading to a blunted effector T-cell immunity and the promotion of tumor growth[9,33,34]. JHU083-mediated glutamine antagonism reduces immunosuppression in the tumor microenvironment

by enhancing the recruitment of long-lived proliferating T-cells, thereby inhibiting tumor growth[6,7]. The antitumor effects of JHU083 have been demonstrated in multiple tumor models including glioma, colon, lung, and triple-negative breast cancers[6,7,11,35]. A prodrug closely related to JHU083, DRP104 (Sirpiglenastat) is in early-phase clinical trials for solid tumors (ClinicalTrials.gov Identifier: NCT04471415)[11]. In addition, DRP104 already has received US FDA Fast Track designation for the treatment of non-small cell lung cancer patients.

In contrast to cancer, the literature on the importance of host glutamine metabolism in infectious diseases such as TB is limited. Specifically, there is no consensus on the impact of exogenous glutamine levels to reduce the progression of a constellation of respiratory

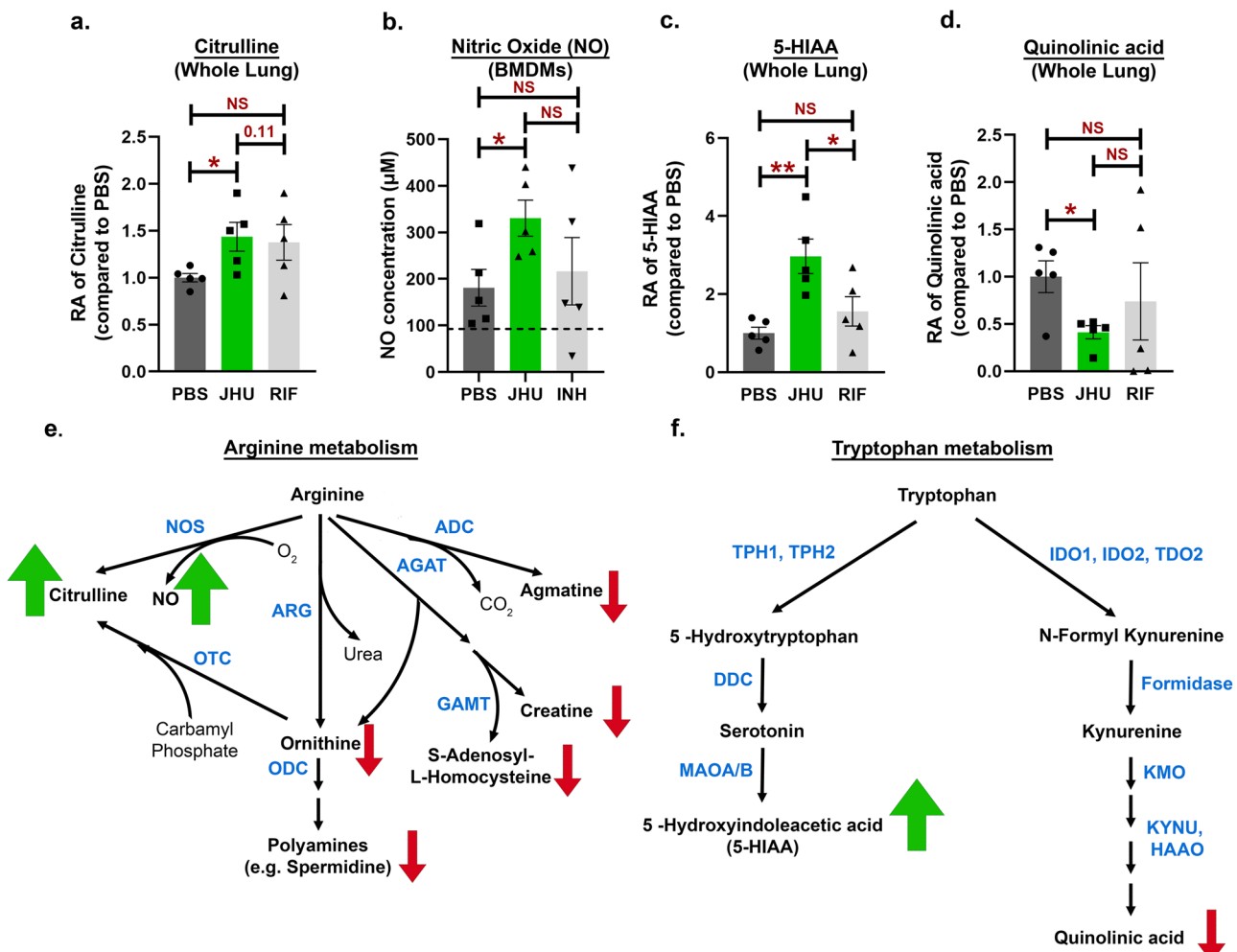

**Fig. 6 | JHU083 treatment drives metabolic reprogramming in *Mtb*-infected lungs.** As described in Fig. 2a, *Mtb*-infected 129S2 mice were treated with JHU083, RIF, or PBS every day starting at day 1 post-infection. Mice were sacrificed at week 2, the lungs were harvested, and total metabolites were methanol (MeOH) extracted as described in the "Methods". The total metabolites were normalized to the tissue weight and then to the untreated controls. We detected significant changes in the levels of (**a**) citrulline in whole lung MeOH extract (*n* = 5/group), (**b**) nitric oxide (NO) in *Mtb*-infected BMDMs treated with JHU083 as measured by the Griess colorimetric assay (*n* = 5/group), (**c**) 5-hydroxyindole acetic acid (5-HIAA) in whole lung MeOH extract (*n* = 5/group), and (**d**) quinolinic acid in whole lung MeOH extract (*n* = 5/group). Schematic representation of the (**e**) arginine and (**f**) tryptophan metabolic pathways with metabolites indicated in black and enzymes in blue. Green/red arrows next to a metabolite indicate statistically significant accumulation/depletion, respectively, in the JHU083-treated whole lung MeOH extract group compared to that for the PBS control. Abbreviations: arginine decarboxylase (ADC), L-arginine:glycine amidinotransferase (AGAT), arginase (ARG), guanidinoacetate N-methyltransferase (GAMT), nitric oxide synthase (NOS), ornithine decarboxylase (ODC), ornithine decarboxylase (OTC), dopa decarboxylase (DDC), 3-hydroxyanthranilate 3,4-dioxygenase (HAAO); indolea-mine 2,3-dioxygenase (IDO), kynurenine 3-monooxygenase (KMO), kynureninase (KYNU), monoamine oxidase A/B (MAOA/B), tryptophan 2,3-dioxygenase (TDO2), tryptophan hydroxylase (TPH1/2). Data were plotted as mean ± SEM. Statistical significance was calculated using a two-tailed student *t*-test considering unequal distribution. The exact *p*-values are provided in the Source Data file. * < 0.05, ** < 0.01. NS stands for non-significant change, *p*-value was >0.05. The experiment was performed twice.

diseases[36]. In the field of acute lung injury, a few studies have shown a beneficial effect of Gln supplementation[37,38], while others have deemed Gln inhibition to be important for preventing progression following injury[39]. In the case of TB, *Mtb* infection has been shown to induce glutamine metabolism transcripts[40,41]. Several studies have found that glutamine metabolism is crucial for T-cell cytokine production, M1 polarization of macrophages, and that glutamine serves as an important carbon and nitrogen source in *Mtb*-infected macrophages[40,42,43]. Thus, while previous reports have described potential roles for host glutamine metabolism during TB, direct inhibition of host glutamine metabolism as a potential host-directed therapy for TB has not been explored previously.

Interestingly, *Mtb* glutamine synthetases have been extensively studied as antibacterial drug targets. Glutamine synthetase catalyzes the amidation of glutamate to glutamine in an ATP-dependent reaction[17]. *Mtb* possess four such enzymes, yet only GlnA1 has been shown to be secreted and to be essential for in vitro and in vivo growth of *Mtb*[16]. In one study, a GlnA1 peptide was found in the serum of 82% latent TB patients[44]. GlnA1 inhibition using methionine sulfoximine (MSO), an inhibitor of glutamine synthases, lowered lung bacillary burden in *Mtb*-infected guinea pigs[15,45]. However, the initial enthusiasm regarding GlnA1 as a potential therapeutic target for *Mtb* infection was dampened by the rapid emergence of spontaneous mutants resistant to MSO (MSO^R). The mutations were found to be in the upstream promoter region of *glnA1* causing GlnA1 overexpression potentially conferring drug resistance[19]. Unlike MSO, DON not only selectively blocks glutamine synthetases but also all glutamine-utilizing enzymes, which include cytidine triphosphate synthase, carbamoyl phosphate synthase, guanosine monophosphate synthase, phosphoribosyl formylglycinamidine synthetase, nicotinamide adenine dinucleotide

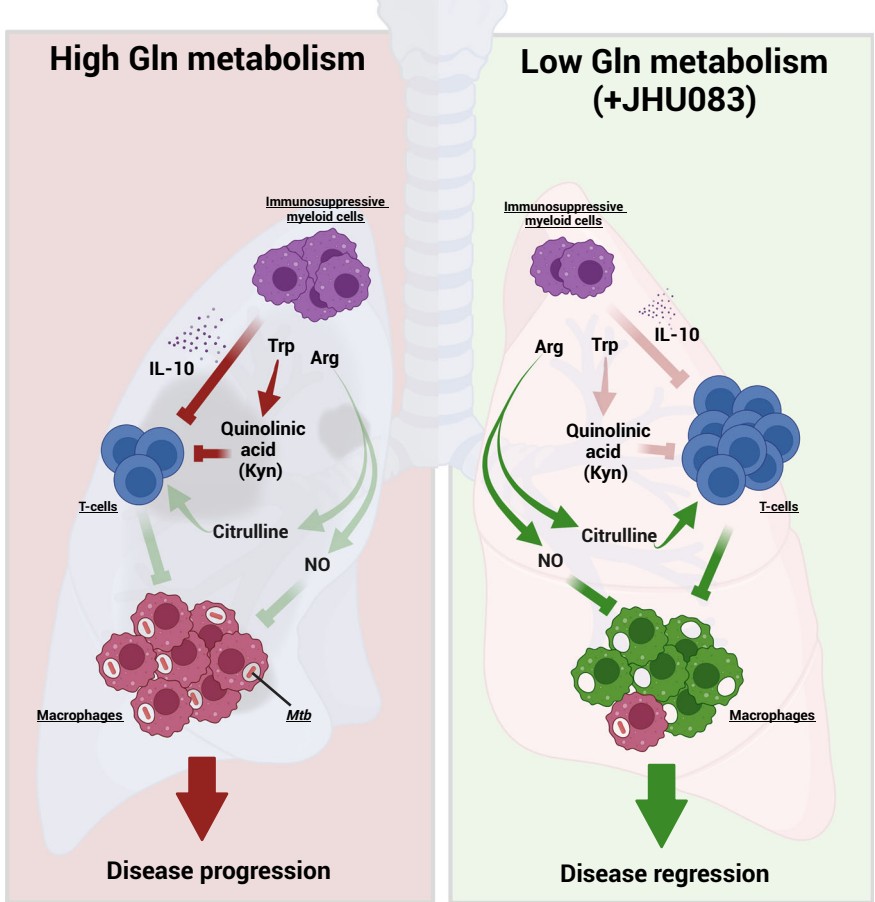

**Fig. 7 | Model depicting the mechanism of action of JHU083.** Under glutamine-sufficient conditions, IL-10 produced by immunosuppressive myeloid cells (MDSCs) and quinolinic acid (a product of the tryptophan/kynurenine pathway) inhibit T-cell proliferation and function, promoting infection and disease progression. Glutamine antagonist JHU083, decreases IL-10-producing MDSCs leading to a higher frequency of activated T-cells. JHU083 treatment was associated with lower lung levels of quinolinic acid suggesting a corresponding reduction in the immunosuppressive metabolite kynurenine (Kyn). It also led to higher lung levels of citrulline suggesting increased conversion of arginine to NO and citrulline. Experiments with *Mtb*-infected macrophages treated with JHU083 confirmed elevated release of NO, well-known to have antimycobacterial activity. These immunometabolic changes then lead to disease regression and improved lung histology. Red lines represent host-deleterious processes, while green lines are for host-protective processes. Trp Tryptophan, Arg Arginine, NO Nitric Oxide. The schematic was created using Biorender.com (Tornonto, Canada) by SP.

synthase, phosphoribosyl pyrophosphate aminotransferase, asparagine synthase and, glutaminases[12]. Consistent with the pleiotropic effect of JHU083 on all Gln-utilizing enzymes, Gln supplementation (5-fold molar excess) fails to rescue the antibacterial activity of JHU083 (Fig. 1d). JHU083 also exhibits potent antibacterial activity ex vivo in BMDM infection models despite a high Gln concentration (2–6 mM) in the cell culture media (Figs. 1f and S1a).

As high glutamine metabolism promotes immune tolerance and immunosuppression in several disease models[7,46], we hypothesized that *Mtb*-associated glutamine release is an immunometabolic virulence mechanism that may be countered by a pharmacologic inhibition of local glutamine metabolism. Data presented here support the hypothesis that the inhibition of Gln metabolism has immunomodulatory effects as shown by enhanced levels of effector T-cells, decreased levels of immunosuppressive myeloid cells, and the absence of a JHU083 protective effect in SCID mice. We also performed whole lung metabolomics in the presence and absence of JHU083 and observed that JHU083-treatment elicits metabolic shifts consistent with enhanced anti-*Mtb* host immune responses.

The flow analysis demonstrated the accumulation of naïve and follicular helper T-cells in JHU083-treated lungs (Fig. 4). The exact mechanism of how these T-cell subsets contain TB needs further exploration. Wolf et al. have demonstrated that naïve T-cells, despite being quiescent, can mount an effective and rapid immune response following activation[47]. Additionally, Orlando et al. have identified a population of naïve CD4+ T-cell population that rapidly produces multiple cytokines in the blood of active TB patients[48]. And more recently, Daniel et al. have demonstrated that homing of naïve lymphocytes to the lung-draining lymph nodes aids anti-TB T-cell response in mice[49]. These studies support the notion that naïve T-cells may contribute to TB containment. Regarding the follicular helper T-cells, a recent report from Dr. Khader's group has demonstrated that the antigen-specific B-cells strategically help localize follicular helper T-cells in lymphoid follicles that aid in *Mtb* containment[50]. Further studies will be required to ascertain the role of the specific B and T-cell subsets in TB progression in the context of glutamine metabolism inhibition.

CD4 + T-cells are critical in driving anti-*Mtb* immune responses. T-cells rely on several amino acids (arginine, tryptophan, glutamine etc.) for their proliferation, activation, and optimal function[51]. Immunosuppressive cells (such as MDSCs and Tregs) create an artificial local depletion of these amino acids, which adversely affects T-cell proliferation and function[14,52]. For example, immunosuppressive myeloid cells overexpress arginase, scavenging arginine and converting it to

ornithine and urea[53]. We found that JHU083 treatment lowered ornithine levels while increasing the level of citrulline in the lung (Fig. 6a). Citrulline originates from arginine in the presence of inducible nitric oxide synthase (iNOS) and generates nitric oxide (NO). We have shown that JHU083 treatment results in elevated levels of NO in macrophages (Fig. 6b). NO has potent anti-mycobacterial activity controlling *Mtb* proliferation in macrophages[54,55]. Citrulline also serves as an arginine reservoir and contributes to maintaining optimal NO concentration when free arginine is scarce[56,57]. In addition, citrulline is also known to augment T-cell proliferation and functions[58]. Diminished levels of other arginine metabolites such as spermidine, creatine, agmatine are also suggestive of a preferential upregulation of the NOS pathway (Fig. S18a).

Another crucial amino acid for T-cell division especially during activation is tryptophan[51]. Suppressive myeloid cells starve T-cells of tryptophan by overexpressing IDO1 and by converting tryptophan to kynurenine[46]. Upregulated IDO expression in tuberculous granulomas has been shown to promote *Mtb* survival in the host, and IDO1 inhibition improves disease outcome[59–61]. Kynurenine, the product of IDO activity, has been shown to accumulate in the plasma of TB patients[62,63]. Kynurenine metabolism also generates quinolinic acid, a potent neurotoxin (if accumulated), which is essential for the de novo synthesis of $NAD^+$[64]. JHU083-treatment lowered quinolinic acid levels in the lungs without decreasing cellular $NAD^+$ levels (Supplementary Data 1). The lowered quinolinic acid levels may be due to either downregulation of tryptophan-kynurenine pathway or heightened rate of quinolinic acid utilization. Interestingly, JHU083 treatment led to accumulation of 5-HIAA, another tryptophan metabolite. 5-HIAA is the primary breakdown product of serotonin and is often used as a proxy for the serotonin measurement. Peripheral serotonin that accounts for >90% of the pool has been shown to exert immunomodulatory effects[65]. Studies have shown that stimulation or activation of serotonin receptor A promotes T-cell proliferation and phagocytic abilities of macrophages[65]. We are not aware of any study reporting the correlation of 5-HIAA with TB.

There are several similarities in the immunomodulatory activity of JHU083 in both the TB and cancer models. JHU083 treatment increased the frequency of CD4 + T-cells and enhanced naive T-cell signatures in both models[6,7]. We also found an increased frequency of follicular helper T-cells and mature B-cells. In both TB and cancer, JHU083 administration lowered the abundance of immunosuppressive myeloid cells while increasing the abundance of myeloid cells with proinflammatory signatures. In both models, the lungs of JHU083-treated mice showed reduced tryptophan metabolism towards kynurenine pathway. However, there were a few key differences between the TB and cancer data. First, we did not observe a decline in the overall frequency of MDSCs in JHU083-treated *Mtb*-infected mice. The reduction that we observed was limited to the IL-10-producing MDSCs population. Second, the whole lungs of JHU083-treated *Mtb*-infected mice showed a trend toward diminished glutamine pools while the cancer data showed an accumulation of glutamine in the tumor microenvironment as measured in excised syngeneic, solid tumors taken from the flank. Indeed, in future studies, it will be interesting to evaluate solid granulomatous lesions themselves, rather than the whole lung, to better understand the effects of JHU083 on the granuloma microenvironment.

Interestingly, unlike the cancer studies[6], we did not observe a statistically significant difference in glutamine levels between control and JHU083-treated lungs. One reason for this may be that the cancer studies measured glutamine levels in heterotopic flank tumors while we evaluated whole lung. JHU083 was originally selected for further study because it was found to be preferentially metabolized from prodrug to DON in the tumor microenvironment[8,12]. Following JHU083 treatment we found rather low levels of DON in lungs (0.856 nmol/g) while its Cmax in murine MC38 tumors (5.38 nmol/g) and murine

plasma (4.1 nmol/ml) is considerably higher[6]. Thus, it is possible that the JHU083-mediated immune cell shifts we observed in lung reflect the action of JHU083 on lymphoid and myeloid cells in other compartments such as blood, spleen or bone marrow with subsequent migration of cells to the lung and that Gln levels in the lung itself do not play a causal role. Another possibility is that JHU083 affects Gln levels in particular lung cell subsets which we did not detect by measuring Gln in the whole lung homogenate MeOH extract. It should also be noted that there is heavy reliance of lung tissue on the de novo Gln synthesis from glutamate and ammonia[66,67], and rodent lungs have also been shown to accumulate as much Gln as skeletal muscle the tissue with the highest free Gln concentration[68]. These factors indicating high Gln pools in the lung, along with the relatively low level of DON we observed, support the concept that JHU083-mediated lung immune cell reprogramming may occur outside of the lung.

While we have demonstrated that JHU083 treatment results in both antibacterial and immunomodulatory activities, we propose that the host-directed activity results in the predominant therapeutic effect. In support of this hypothesis, we have shown that (1) JHU083 loses its therapeutic efficacy in *Mtb*-infected immunocompromised mice (Fig. 3), that (2) there are significant beneficial changes in lymphoid and myeloid cell populations with JHU083, and that (3) these immune cell changes mostly occur at week 2 post-infection and treatment when there is no difference in the lung bacillary burden between the JHU083- and PBS-treated groups (Fig. 2b). Additionally, the pharmacokinetic data demonstrates that the DON drug level in *Mtb*-infected lungs are at least 10,000-fold lower than the MIC value of DON/JHU083 (Fig. 2c) further supporting a host-directed effect. To further delineate antibacterial vs immunomodulatory activities of JHU083, evaluating efficacy of JHU083 in mice infected with an *Mtb* mutant resistant to JHU083 (JHU^R) would be valuable, and the lack of such data are a limitation of this study. We made numerous unsuccessful attempts to isolate resistant mutants on JHU083-containing agar plates although MSO^R mutants were readily found by the same method. Another limitation is that the metabolomic data presented in Figs. 6 and S16–S18 are the result of an unbiased, untargeted analysis of ~150 cellular metabolites from the whole lungs of *Mtb*-infected, JHU083-treated, and untreated mice. Future studies to determine the metabolic status of specific immune cell populations and to measure metabolite fluxes with and without drug treatment will be valuable in refining our current observations. Additionally, for all the experiments presented in the manuscript, JHU083 treatment was initiated one-day post-infection to provide this experimental drug the maximum chance to display a therapeutic benefit. However, testing JHU083 in acute and chronic models of TB infection either as a monotherapy or in combination with the standard chemotherapy regimen is crucial to ascertain its potential as an immunotherapeutic for TB.

Based on both the results presented in the prior literature by Horwitz and colleagues[15–17] and in this study, we propose that *Mtb* lung infection generates local microenvironments with elevated glutamine levels. This in turn leads to accumulation of immunosuppressive myeloid cells, reduced effector T-cell function, and downregulation of NO and citrulline synthesis. We have shown that administration of JHU083 in the TB mouse model results in the decreased level of immunosuppressive myeloid cells, enhanced levels of effector T-cells, and increased levels of citrulline and NO production (Fig. 7). Lastly, while JHU083 has both a direct antibacterial effect and immunomodulatory activity, the kinetics of its effects and its lack of efficacy in SCID mice suggest that JHU083 functions predominately as a host-directed immunotherapeutic in the mouse model of TB.

## Methods
### Ethics regulation
All animal studies were performed per the protocols approved by the Johns Hopkins Animal Care and Use Committee of the Johns Hopkins

School of Medicine. All animals were procured and handled per the approved mouse protocol number M022M466. All animals were housed in individually ventilated cages under 12 h dark/light cycle in a room with ambient temperature between 20 and 24 °C and a relative humidity of 45–65%. Each cage contained no more than five mice of the same strain. Mouse euthanasia was performed in the most humane and painless way possible.

### Animal infection studies

Six to ten weeks old 129S2 female mice ($n = 4$–10/group) were procured from (Charles River Laboratories, Wilmington, Massachusetts). Six to ten weeks old female C3H ($n = 5$/group), C3HeB/FeJ mice ($n = 6$/group) and Balb/c SCID ($n = 5$–10/group) were procured from Jackson Laboratory (Bar Harbor, MA), respectively. 200–300 CFU of *Mtb* H37Rv strain was used for all infections. For Balb/c SCID infection studies, ~40 CFU of *Mtb* H37Rv strain was used. The Bishai lab maintains frozen stocks of *Mtb* H37Rv that are sequenced-verified regularly. The day after infection, we randomly distributed mice into three groups: (1) PBS, (2) JHU083, and (3) RIF group. For JHU083, either of the two dosing regimens was used; (1) Daily (1 mg/kg dose per day for first week, followed by 0.3 mg/kg daily) and (2) Alternate (1 mg/kg dose per day for first week, followed by 1 mg/kg on Mon, Wed and Fri). 12.5 mg/kg RIF was administered daily as positive control. All the drugs were administered orally. Only female mice were used for the studies for easier co-housing and to keep the managable number of animals per group.

For CFU enumeration, lungs were collected at weeks 2 and 5 post-infection and treatment. The lungs were then homogenized, serially diluted in PBS, and 100 µl aliquots spread on 7H11 selective plates. For the survival study, ten mice per group were kept under daily observation. For the body weight measurements, five mice from each group were weighed weekly. C3HeB/FeJ mice infection studies were terminated early as the untreated mice tended to die ~4.5 weeks post infection.

### MIC and MBC determination

Alamar Blue Assay for MIC determination was performed in 96-well clear-bottom microplates as described earlier[69,70]. Stock solutions of JHU083, DON MSO, and RIF were diluted 2-fold in 0.1 ml 7H9 broth. 0.1 ml of *Mtb* H37Rv grown to mid-exponential phase was diluted and added to each well and incubated at 37 °C. After 7 days, 20 µl of 10X Alamar Blue solution (Invitrogen) along with 12.5 µl of Tween 80 (20%) were added to the well. After overnight incubation (>16 h), fluorescence intensity was measured using a fluorescence microplate reader at 544Ex/590Em nm. Minimum inhibitory concentration (MIC) was defined as the lowest drug concentration at which 90% growth inhibition was observed. RIF and no drug wells were used as positive and negative controls. The media used was 7H9 broth without Tween-80 addition.

$10^5$ bacteria were seeded in 2.5 ml 7H9 media in 15 ml polystyrene tubes. Varying concentration of JHU083 in two-fold increments (0.125 µg/ml to 64 µg/ml) was added on day 0. After two weeks, the tubes with no visible growth were assigned as "zero growth". The culture from zero tubes were serially diluted and plated on 7H11 plates. Following 21 days incubation at 37 °C, the colonies were counted and CFU was enumerated. Three tubes each with (1) media containing no inoculum and, (2) media containing inoculum but no drug, were used as blank and growth controls, respectively. The lowest drug concentration at which zero growth was observed was defined as the MIC. MBC was defined as the lowest drug concentration that caused >2 $\log_{10}$ CFU/ml reduction compared to the initial inoculum.

### Bone-marrow-derived macrophage infection studies

BMDMs were isolated, as described earlier[71]. Briefly, BMDMs were isolated from the femurs of 3–4 months old female C57BL/6 (Jackson Laboratory, Bar Harbor, MA) or 129S2 mice (Charles River Laboratories, Wilmington, Massachusetts). Generally, 10 mice were sacrificed for one BMDM experiment. The isolated monocytes were differentiated into macrophages in RPMI medium-Glutamax (Gibco; 61870-036) containing 10% FBS (Gibco; 16140071) and 1X antibiotic antimycotic Solution (Sigma Aldrich; A5955) and 30% (v/v) L929 conditioned media. 50,000 BMDMs were polarized using 10 ng/ml IFNγ per well for 24 h. The macrophages were then infected with *Mtb* H37Rv at an MOI of 2 for 4 h. The cells were washed with prewarmed media and incubated with 200 µg/ml gentamicin to eliminate the extracellular bacteria. The cells were treated with DON (1X or 10X MIC), JHU083 (10X MIC) and INH (32X MIC). The cells were then collected on days 1, 3, and 5, lysed with 0.25% SDS, diluted, and then plated on 7H11 selective plates. The CFU enumeration was performed on days 21–28, and the final values were plotted on a log scale of 10.

For testing the direct cytotoxic activity of DON upon macrophages, 50,000 BMDMs were plated in triplicates in 200 µl volume in a 96-well plate. The cells were then pretreated with 10 ng/ml IFNγ per well. After 24 h, DON (1X and 10X MIC daily) and INH (32X MIC on day 1) were added. On days 1, 3, and 5 post-drug treatment, 3-(4,5-dimethylthiazol-2-yl)-5-(3-carboxymethoxyphenyl)-2-(4-sulfophenyl)-2H-tetrazolium, inner salt; MTS reagent (Promega) was added, and the absorbance was recorded at 490 nM with a Microplate reader (Bio-Rad, Hercules, CA). The absorbance from the wells with no cells and only media was subtracted to negate the background signal.

### Lung histopathology estimation

For the histology, left lung lobes were harvested from PBS- and JHU083-treated 129S2 ($n = 3$/group) and C3HeB/FeJ ($n = 6$/group) mice and stored in formalin. The time-point was 4.5- and 5-weeks post infection/treatment for C3HeB/FeJ mice and 129S2, respectively. While necrotic granulomas usually require 6 weeks to develop after low-dose challenge, in our experiment with 200–300 CFU as the challenge dose, necrotic granulomas were present in the control mice at the 4.5 week time point. The formalin-fixed tissues were then sectioned and stained with Hematoxylin and Eosin (H&E). Stained sections were imaged at 40X magnification. The granulomas or tuberculous lesions were defined as an aggregation of epithelioid macrophages which may also contain giant cells and may or may not be surrounded by a cuff of lymphocytes. The lesion areas were quantified using ImageScope Software (Leica Biosystems) and plotted using GradPad Prism software. Percent lesion area = (total lesion area X 100)/total lung area

### Single-cell suspension preparation

For multicolor flow cytometry analysis, lungs and spleens were collected in 5 ml MACS Tissue storage solution (Cat: 130-100-008; Miltenyi Biotec, Gaithersburg, MD) followed by storage at 4 °C until processing. The lungs were dissected into individual lobes, and single-cell suspension was prepared using the mouse lung dissociation kit (Cat: 130-095-927; Miltenyi Biotec, Gaithersburg, MD) and Gentle-MACS™ Dissociator following the manufacturer's protocol. Spleens were mechanically dissociated in digestion buffer (RPMI 1640 + 10% FBS + 0.2 mg/ml collagenase D + DNAse). Both lung and spleen cells were then incubated with ACK lysis buffer at RT for 2–5 min to lyse the red blood cells. The cell suspension was then washed with RPMI complete media and resuspended in the appropriate volume of the same media. Trypan blue staining was performed to assess the viability of the cell suspensions.

### Multicolor flow cytometry

Lung single-cell suspensions were incubated with TruStain FcX™ (anti-mouse CD16/32) antibody (Cat: 101320; BioLegend, San Diego, CA) in eBioscience™ Flow Cytometry Staining Buffer (Cat: 00-4222-57, San Diego, CA) for 20 min at room temperature to block non-specific antibody binding. The cells were then incubated with appropriate

antibody cocktails and fixation buffer (Cat: 420801; BioLegend). For intracellular staining, the cells were stained using True-Nuclear™ Transcription Factor Buffer Set (Cat: 424401; Biolegend) following the manufacturer's protocol. Intracellular cytokine stimulation and staining was performed as described earlier[72]. Briefly, lung cells were incubated with cell activation cocktail (Biolegend) and monensin for 4 h. The cytokine staining was performed using Cyto-Fast™ Fix/Perm Buffer Set (Cat: 426803; BioLegend) following the manufacturer's protocol. The stained cells were stored in Cyto-Last™ Buffer (Cat: 422501; BioLegend) at 4 °C till the data acquisition. The data were acquired on BD LSRFortessa™ Cell Analyzer (BD Biosciences, San Jose, CA) and the data were analyzed using FlowJo version 10.8.1 (Tree Star). All flow antibodies were titrated to identify the concentration with maximum specificity coupled with minimum possible spillover. The gating strategy was defined using single-stain and FMO controls (for low-expression markers). Gating strategies are provided in the Supplementary Figs. S6, S7, S8, S12 and S13. Instead of beads, splenocytes were used for single-cell stain controls and for creating compensation matrices. All flow antibodies were titrated to identify the optimal concentrations for staining protocols. Only flow antibodies validated for specificity by the manufacturers were used.

## Pharmacokinetics study

Six to 10 weeks old female 129S2 mice ($n = 5$/group; Charles River Laboratories, Wilmington, Massachusetts) were infected with ~200 CFU *Mtb* H37Rv. The day after infection, we randomly distributed mice into two groups: (1) PBS and (2) JHU083. For oral administration of JHU083, alternate dosing regimen (1 mg/kg dose per day for first week, followed by 1 mg/kg on Mon, Wed and Fri) was followed. After two weeks, either PBS or 1 mg/Kg JHU083 was administered. The mice were sacrificed within 30–40 min of the dosing, lungs were harvested, weighed and flash frozen in liquid nitrogen. 500 µl of 80% HPLC-grade methanol: water (v/v) (cooled to −80 °C) was added to frozen tissue piece(s). Bead beating for 30 s x 4 times was performed using CK14 tubes and Percellys Homogenizer (Bertin technologies). The rest of the extraction was performed as described in the "Metabolite Extraction" section. The concentration of the drug was normalized to the weight of the lung tissue used for each sample.

## Metabolite extraction

For LC/MS based metabolomics, lung tissues were collected from all three groups and flash-frozen in liquid nitrogen followed by storage at −80 until further processing. For the processing, 10–30 mg lungs were homogenized in 500 µL of methanol:water (80:20, v/v) extraction solution, vortexed and centrifuged at 14,000 x *g* for 10 min at 4 °C. The supernatant fluid was stored at −80 °C overnight to precipitate proteins. The supernatant was centrifuged at 14,000 x *g* at 4 °C and filtered through 0.22 µm acetate filters (Corning, Corning, NY; Cat: 8160), lyophilized and stored at −80 °C for subsequent analysis.

## Metabolite measurement with LC-MS

The dried metabolite extracts were resuspended in 50% acetonitrile solution. An LC-MS system consisting of an Agilent 1290 Infinity II Binary UHPLC pump and a Bruker timsTOF Pro II mass spectrometer was used for the LC-MS based metabolomics profiling. HILIC-LC chromatographic separations were performed on the above UHPLC using a Waters XBridge BEH Amide column (2.1 × 150 mm, 1.7 µm). The LC parameters were as follows: autosampler temperature, 4 °C; injection volume, 2 µl; column temperature, 40 °C; and flow rate, 0.20 ml/min. The solvents and optimized gradient conditions for LC were: Solvent A, Water with 0.1% formic acid; Solvent B, Acetonitrile with 0.1% formic acid; A non-linear gradient from 99% B to 45% B in 25 min with 5 min of post-run time. MS spectra were collected using a timsTOF Pro II mass spectrometer (Bruker Daltonics) equipped with

IonBooster ESI source. The mass spectrometer was operated in negative mode with auto MS/MS method. The optimized operation parameters were End Plate Offset, 400 V; Capillary Voltage, 1000 V; Charging Voltage, 300 V; Nebulizer Pressure, 4.1 bar; Dry Gas 3 L/min; Dry Gas Temperature, 200 °C; Vaporizer Temperature, 350 °C; The mass scan range, 70–1100 m/z; Scan rate, 12 Hz. Spectra were internally mass calibrated at the beginning of every sample run by infusion of a small fragment of reference mass solution using a syringe pump connected to the sprayer feeding into the ESI source. Data were acquired with Compass HyStar 5.1 acquisition software and processed with TASQ 2022. The analyte database used for metabolite identification was developed in house with retention times based on the HILIC method. We note that our methods do not distinguish some metabolites with the same formula and very similar structures, hence care should be taken in interpretation of such data.

## Colorimetric nitric oxide assay

BMDMs were isolated from 3 to 4 months old 129S2 mice (Charles River Laboratories, Wilmington, Massachusetts), IFNγ-activated, infected with *Mtb* H37Rv and treated as described in the "*Bone-marrow-derived macrophage infection studies*" section. 24-h post drug treatment, supernatant fluid was collected and centrifuged at the maximum speed. NO was measured in the clarified supernatant fluid as per the manufacturer's instructions (Abcam, Ab65328). Uninfected BMDMs seeded at the same density were used as the control. 0.5 million BMDMs were seeded per well in 24 well plates (5 wells per group).

## Western blotting for IDO1 quantitation

Lung tissues were transferred into 500 µl of T-PER™ Tissue Protein Extraction Reagent (Thermofisher Scientific; Cat: 78510) containing 1X Halt Protease Inhibitor Cocktail Thermo Scientific; Catalog: 87785. The tissues were homogenized by bead beating for 30 s x 4 times using CK14 tubes and Percellys Homogenizer (Bertin technologies). The protein concentration was estimated using Pierce™ BCA protein assay kit following manufacturer's instructions (ThermoFisher Scientific, Waltham, MA, USA). An equivalent of 10 µg protein was loaded per lane onto SDS-PAGE gel, electrophoresed, transferred to PVDF membrane. The membrane was blocked with 5% milk at RT. After 2 h, the membrane was incubated with 1:1000 dilution of α-IDO1 antibody (Cell Signaling Technology; Catalog: 51851 S) overnight at 4 °C, washed and then incubated with HRP-conjugated anti-rabbit secondary antibody (Cell Signaling Technology; Catalog: 7074 S). After IDO1-imaging, antibody stripping was performed using Restore™ PLUS Western Blot Stripping Buffer (Thermofisher Scientific; Cat: 46430). The blot was then probed with HRP-conjugated β-actin antibody (Cell sigaling technology; Catalog: 5125 S). Proteins were detected using Pierce ECL substrate and imaged with KwikQuant Imager (Kinde Biosciences, LLC). Densitometric quantitation of the bands was performed using ImageJ software (NIH). Only antibodies validated for specificity by the manufacturers were used.

## Statistics and reproducibility

No statistical method was used to predetermine sample size. All the experiments were randomized. All data collected is reported in the manuscript. For animal experiments, premature death of the mice resulted in decreased sample size occasionally. No data were excluded from the analyses. After aerosol infection, animals were randomly assigned to individual groups. The investigators were not blinded to allocation during experiments and outcome assessment.

## Reporting summary

Further information on research design is available in the Nature Portfolio Reporting Summary linked to this article.

## Data availability

The raw data generated during the study has been provided as a separate MS Excel spreadsheet labeled "Source Data file". Metabolomics dataset has been provided as "Supplementary Data 1". Any queries regarding the data should be addressed to the corresponding author. Source data are provided with this paper.

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

## Acknowledgements

We gratefully acknowledge the support of NIH grants AI155602, R01CA226765 and that of the Bloomberg-Kimmel Institute for Cancer Immunotherapy. Flow cytometry was performed at the SKCCC flow/mass cytometry core. Lung histology was performed at SKCCC oncology tissue services core. We are grateful to Drs. Marcus Horwitz and Michael Tullius for graciously sharing GlnA1 auxotrophic mutant strains. We acknowledge past and present members of the Bishai lab for helpful suggestions and discussion throughout the study.

## Author contributions

S.P., J.R.M. and W.R.B. conceptualized the study and designed the research approach; B.S.S., R.R., R.D.L., J.D.P. provided inputs critical to the study; B.S.S. and R.R. provided JHU083 for the study; S.P., J.S., S.L., B.K. performed animal studies; S.L. performed MIC and MBC determination assays; S.P., J.S. and L.Z. performed metabolomics study; S.P. performed flow cytometry; S.P. and J.A. performed pharmacokinetics study. S.P., J.R.M. and W.R.B. analyzed data; S.P., J.RM. and W.R.B. wrote the paper; S.P., J.S., S.L., L.Z., S.A., B.K., R.D.L., R.R., J.D.P., J.R.M., B.S.S. and W.R.B. critically reviewed the manuscript.

## Competing interests

S.P., J.S., S.L., L.Z., B.K., J.R.M. and W.R.B. declare no conflict of interest. J.A., R.R., J.D.P. and B.S.S. are inventors on multiple Johns Hopkins

University (JHU) patents covering glutamine antagonist prodrugs including JHU083 and their utility. These patents have been licensed to Dracen Pharmaceuticals Inc. R.R., J.D.P. and B.S.S. are founders of and hold equity in Dracen Pharmaceuticals Inc. This arrangement has been reviewed and approved by the JHU in accordance with its conflict-of-interest policies. R.D.L. is an inventor on US patent 10842763 submitted by Johns Hopkins University and licensed to Dracen Pharmaceuticals that covers the use of glutamine analogues, such as JHU083 (DRP-083), for cancer immunotherapy. The authors declare no other competing interests.

## Additional information

[1]Center for Tuberculosis Research, Department of Medicine, Johns Hopkins School of Medicine, Baltimore, MD, USA. [2]Department of Oncology, Johns Hopkins School of Medicine, Baltimore, MD, USA. [3]Johns Hopkins University, Baltimore, MD, USA. [4]The Bloomberg-Kimmel Institute for Cancer Immunotherapy, Johns Hopkins School of Medicine, Baltimore, MD, USA. [5]Johns Hopkins Drug Discovery, Johns Hopkins School of Medicine, Baltimore, MD, USA. [6]Department of Neurology, Johns Hopkins School of Medicine, Baltimore, MD, USA. [7]Department of Pharmacology and Molecular Sciences, Johns Hopkins School of Medicine, Baltimore, MD, USA. [8]Department of Neuroscience, Johns Hopkins School of Medicine, Baltimore, MD, USA. [9]Department of Psychiatry and Behavioral Sciences, Johns Hopkins School of Medicine, Baltimore, MD, USA. [10]Present address: Calico, South San Francisco, CA, USA.
✉e-mail: wbishai1@jhmi.edu

