## [Peer Review File · Nature Communications]

Glutamine metabolism inhibition has dual immunomodulatory and antibacterial activities against *Mycobacterium tuberculosis*REVIEWER COMMENTS

Reviewer #1 (Remarks to the Author):

The manuscript by Parveen et al. highlights the role of a known glutamine metabolism antagonist, JHU083 in inhibiting Mtb proliferation and growth in vitro and in vivo. The authors propose that JHU083-mediated glutamine metabolism inhibition results in dual antibacterial and host-directed activity against tuberculosis.

Major comments

Whereas the role of glutamine metabolism in T cell activation is described in the literature, the use of the glutamine metabolism inhibitor JHU083 resulting in T cell activation during Mtb infection is very interesting and relevant to the discovery and development of new HDTs for the control of TB. The study is well written, focused and the extensive flow cytometry data sets are appreciated. The biggest concerns include the use of selective representation of data sets from different animal models, and the lack of mechanism. For example, in case of the former, DON was used in the in vitro CFU assay instead of JHU083, BMDM from C57BL/6 mice were used whereas 129S2 mice were used for in vivo experiments, and C3HeB/FeJ mice were used to demonstrate histopathology. Also, different time points were used for CFU determination in 129S2 (2 and 5 week) and C3HeB/FeJ mice (4 weeks). These concerns could potentially be addressed. Lastly, although inhibition of glutamine metabolism was not supported by the metabolomics data (although there appears to be a trend towards significance), and complicates the mechanism, my view is that this is not a critical caveat.

What is the rationale of using 129S2 mice that are typically used for transgenic/knockout model development? Why were C3He or BALB/c strains not used? I see that for CFU assessment, C3HeB/FeJ mice were used in one panel but throughout the manuscript (flow data), 129S2 mice were used. The authors should provide an explanation.

In Figure S3 and S4, the gating strategy looks a bit subjective. What were the controls - FMO, isotype controls for generating the gates? Please include this information. Gates are shown without having a distinct zone and looks marginal. Can the authors comment on these issues perhaps in the figure legends?

Line 42; The authors mentioned reduced glutamine levels in the abstract; since these levels were not statistically significant different from the untreated group, I suggest they remove this statement. This will then be consistent with the Discussion.

I suggest the authors remove the last paragraph from the introduction as it is virtually identical to the abstract with a few extra details.

Figure 1C; can the authors indicate indicated in the legend the number of replicates, independent experiments, standard deviation? Given the significance of the study, the authors should have used a

more rigorous approach for determining Mtb killing; e.g., CFU-based killing assays.

Line 157; it will be helpful if the authors could provide a rationale for initiating treatment one day post infection.

Line 179; SCID mice details are lacking in the Animal Infection Studies in the materials and methods section.

In lines 206 -209, how does increased naive T cells (CD4+ and CD8+ expressing CD44-CD62+) protect treated mice from TB compared to untreated mice? Why are naive CD4 cells (Fig 4E) presented in % whereas naive CD8 cells (Fig 4G) are presented in MFI?

Line 267; given that the reduction in glutamine levels is not statistically significant I suggest the authors change “The most notable changes...” part of the sentence to indicate this fact. I suggest authors concisely state why changes in glutamine levels were not observed.

Line 337; remove the Unpublished Data.

In Fig 4 and Fig 5, the immunology data are inconsistent as some of the cell markers are shown as % (4A, E, F and Fig 5A, B, E, F, G), whereas the other panels were presented in MFI. Why is this the case?

Minor comments:

- Line 167 and line 168 – Fig 2C should be Fig 2D and Fig 2D should be Fig 2C respectively.
- In figure 2E, the lung weight should be in terms of mg, or change the axis accordingly. Weight of lungs cannot be 200g.
- In the method section of the supplementary file, the method for “Lung histopathology estimation” (in line 71) indicate 5 weeks of post-infection, whereas in the result section of the manuscript (line 169-170) it indicates 4 weeks of infection/treatment was used for histopathological examination.
- Fig 3C indicates the SCID mouse body weight was measured for up to 7 wk (i.e., 49 days). However, survival data indicate that both PBS and JHU083-treated mice die before 7 wk time. Is this a mistake in numbering weeks post-infection, instead of 6 it is written as 7 weeks (as each time point is showing 1 wk difference instead of the last time point).
- Line 214 and line 218 - Fig 3I & 3J should be Fig 4I & 4J and Fig 3K & 3L should be Fig 4K & 4L.
- Line 264 and line 372 – there is no “Supplementary Table S1”.

Reviewer #2 (Remarks to the Author):

The work entitled “Glutamine metabolism inhibition has dual immunomodulatory and antibacterial activities against *Mycobacterium tuberculosis*” by Parveen and others details studies testing whether a glutamine metabolism antagonist colloquially named ‘JHU083’, an orally available prodrug of the 6-diazo-5-oxo-L-norleucine, would further reduce immunopathology associated with tuberculosis infection. Administration of JHU083 was tested using aerosol infection in 129S2 mice and rifampin was used as a comparator along approximately the same timeline. Numerous biological endpoints were measured to assess performance of JHU083, and this included bacillary loading in the lung, cell subset-specific activity, inflammatory mediators, global metabolomics (of the whole lung), and pathology spec. to granuloma formation.

There are no major concerns with this work that would completely halt recommendation for publication. However, there are some questions about approach in the general experimental design that are questionable and that must be addressed by the authors before proceeding.

There was little to no time spent on understanding (or commenting on) basic pharmacokinetics of this prodrug. It was clearly stated that there was transformation that took place once in vivo (to 6-diazo-5-oxo-L-norleucine) and the corresponding oral bioavailability, which should have ‘fueled the fire’ to perform a careful pharmacokinetics study prior to a therapeutics evaluation. Yet, studies with 6-diazo-5-oxo-L-norleucine which showed minimal benefit and GI-related ‘difficulties’ was the only mention of anything remotely resembling an idea about pharmacokinetics. What about the esterases that transformed the prodrug? Was this measured in the context of dosing, and how variable is the in vivo concentrations in the context of oral dosing? Is there any difference in another route of administration that may effect transformation?

Only one dose (1 mg/kg) of JHU083 was presented in this work. Was other dosing regiments attempted, and if not, then why would only one dose regiment be used when considering evaluation of any treatment. Also, only one regiment (daily) was used in the evaluation. With virtually no pharmacokinetics data to inform dose or regiment, how was the decision made to arrive at these parameters? Either an incredibly risky or lucky way to plan this study.

This work is a comprehensive and at times overwhelming assessment of the cellular mechanism-based changes associated with introduction of an experimental reagent drug in a murine model of tuberculosis. The amount of endpoint analysis in the context of measured endpoints is borderline unfocused – there is so much endpoint data provided with minimal planning on the basic parameters necessary to determine optimal administration and route relative to kinetics of JHU083 (referral to earlier comments).

Accordingly, there is a concern with organization of the data as it relates to assessment. It seems that there is little regard for omission of some data with no explanation, wherein it (seems) to the reader is to overwhelm with data, rather than provide a careful interpretation of why one or the other measurement was excluded. A clear example is the exclusion of rifampin comparator in Figure 2C. 2D. and 2G.; body wt gain, time to death, and % granulomas, three otherwise major indicators that were excluded from graphs, yet in Fig. 2E lung weight rifampin appears again. Similarly, in Fig 3 that details results with the SCID mice, rifampin appears as a comparator. The immediate concern is why this data for the

comparator was excluded in Fig. 2, but more generally is this thematic for understandably densely packed report. A similar and related concern is the inclusion of analysis that (an opinion) adds minimally to the overall support of the conclusions of this work. A clear example is inclusion of the metabolomics data, that is already discounted by the authorship because of whole lung digestion rather than specific cell subsets, and largely uninterpretable heatmap presentation with relatively weak statements in the legend of the figure (actually no conclusions) and in the text, line 267, “The most notable changes in JHU083-treated lungs were as follows: (i) A ~17% reduction in the glutamine levels (P = 0.17; Fig. 6A)”. The P value provided is nonsignificant, but it is “the most notable change”? It is hard to understand why this would be included in this manuscript with so many other endpoints that show benefit of JHU083 administration.

Very minor

Line 88, ‘track’ should be ‘tract’

Line 336-337, please consider removing any references to ‘unpublished data’ as this is completely unhelpful

Line 374, 5-HIAA is undefined

Fig S2, legend, consider changing ‘two-times’ to ‘twice’

Fig S11, Fig S12, the graphic quality of these two heatmaps are horrible

Reviewer #3 (Remarks to the Author):

Authors within this group have developed a pro-drug for DON which has demonstrated remarkable potency in treating a variety of tumors in mice. These authors have demonstrated that JHU083 affects tumor-infiltrating T cells by increasing expression of markers of activation and proliferation and by decreasing the number and activity of myeloid-derived suppressor cells. In this work, the authors explore the effect of JHU083 on a murine model of tuberculosis. Firstly, based on the known role of Mtb glutamine synthetase in pathogenesis, the authors show that the compound has a direct growth inhibitory effect on the bacterium, but the mechanism is not explored even though this would be easy to do. Next the authors show that treatment of infected mice has an extremely modest effect on bacterial burdens in organs of infected mice although by treating the mice with a sub-optimal dose of Rifampicin, the result seem good. The compound does, however, have a dramatic effect on survival, body weight and lung weight of infected mice suggesting that something important is happening. A non-standard mouse strain is used which makes data interpretation a bit complex and the lack of efficacy in SCID mice is promising although the commonly used immunocompetent control strain is not compared which

makes benchmarking tough. The authors next analyze different cell types in the lung but although statistically measurable differences are seen, there is no clear answer as to the functionally relevant cell population induced by JHU083. In many cases, similar changes are seen for Rif-treated animals. The effect on myeloid derived suppressor cells is suggestive but the more convincing follow-up work reported previously by some of the authors in this group is not done. In contrast to the cancer work where the tumor microenvironment is often hypoxic, the authors do not test the compound in a necrotic granuloma model in mice even though the mouse strain that can develop such granulomas given the right experimental setup is used. Thus, comparing effects in tumor experiments reported in the literature with the effects seen here is impossible. There are differences but some of these are also seen in Rifampicin-treated mice. Metabolites are measured and the most important metabolite that could possibly explain the specific effect of JHU083 is quinolinic acid, but the authors stop short of demonstrating that IDO activity is key to the host-directed effect of JHU083.

In summary, the results are interesting and point to a host-directed benefit of JHU083 in chemotherapy of Mtb-infected mice. A few key experiments would allow the authors to address some of their hypotheses more conclusively.

Specific comments

DON and JHU083 inhibit MTb growth in vitro. If the assumption is that these inhibit the essential GlnA1 enzyme, gln supplementation would rescue (as reported for MSO). Does gln or glu supplementation affect the MIC? Is JHU083 static or cidal against Mtb in vitro?

Is JHU083 converted to DON by Mtb?

The antibacterial efficacy against Mtb growing in macrophages suggests a rather limited effect against Mtb in host cells but as mentioned by the authors, could reflect poor access to the phagosome. Alternatively, the Gln in the tissue culture medium could rescue inhibition.

Fig. 2: mice are treated 1 day after infection which is an extremely early stage of infection. Did the authors attempt treatment at later stages?

Fig. 2: Rif dosing at 1.25mg/kg seems extremely suboptimal. Why did the authors choose this very low dose?

The dosing strategy for JHU083 is similar to that used in Leone et al 2019 and Oh et al. 2019 – for the first 5 days mice are given 1mg/kg but the dose is 3-fold lower later. Is this due to toxicity?

The C3HeB/FeJ mouse model can form necrotic granulomas. However, the necrotic granulomas only develop after a month, usually at lower implantation doses. Thus, stating that the activity of JHU083 was assessed in a mouse model which develops necrotic granulomas is misleading to most readers.

Fig. 2G: granuloma numbers were measured – what did the authors define as a granuloma? This is not mentioned in the M&M.

Lines 167 and 168: reference to the MTD and body weight panel is switched (Fig. 2C versus 2D)

Fig. 3: It's frustrating that the authors don't show the efficacy results for JHU083 in immunocompetent Balb/c mice. The authors routinely use Balb/c mice for published work thus it seems strange that JHU083 efficacy in this mouse strain is not reported. This would be important to benchmark these results.

In all cell analyses, the uninfected mouse control group treated with JHU083 is not shown. This is an important control for this mouse strain.

It is intriguing that the T cells that infiltrate the lungs of JHU083 Mtb infected mice express high levels of markers for naïve T cells and follicular T cells. This response is transient. Thus, it is unclear how this increase in T cells results in the small differences in organ bacterial burdens and dramatic changes in mortality. Fig. 4: Despite many changes being statistically significant, the actual changes are often quite

small (eg. in untreated mice 4% of CD45+ cells are naïve CD4+ cells whereas this increases to maybe 7% in JHU083 treated mice) which further raises questions about which of the observed changes are important for the apparent positive effect of JHU083. Similarly, the monocytic myeloid-derived suppressor cells are statistically different but the % change is small (2% versus 3% at week 2 for untreated vs JHU083-treated mice, respectively). Many of the cellular differences are comparable to that measured in the Rif-treated group. If JHU083 is having specific immunomodulatory effects, one would expect the relevant functionally important differences to be specific for JHU083.

The authors observe lower levels of gln. Previous work had demonstrated increased gln/glu ratios which seemed to make sense when considering the mechanism of JHU083. What were the Gln/glu ratios?

There is a small increase in citrulline, and the authors hypothesize that this could be due to iNOS induction and activity. There are inhibitors available to directly test this although not recommended by this reviewer since Rif treatment has a similar effect. The increased citrulline levels contrasts with the decreases in tumor citrulline levels reported in JHU083 treated mice (Oh et al. 2019).

Oh et al. (2019) reported decreased kynurenine levels in lung tumors of JHU083-treated mice. In this work only quinolinic acid, a potential byproduct of kynurenine metabolism is detected. Was kynurenine not detected?

The model in Fig. 7 suggests that JHU083 is responsible for disease regression by virtue of the lower production of kynurenine and increased production of nitric oxide. The problem with the NO-mediated control aspect of this model is that Rif-treated mice had similar enhanced nitric oxide levels. However, the effect of IDO is rather appealing since IDO activity is a known suppressor of T cell function and enhances the activity of myeloid-derived suppressor cells. It would have been wonderful to see the authors validate this prediction using an IDO inhibitor. Note that IDO inhibitors have been used for host-directed therapy studies for animal models of TB.

POINT BY POINT RESPONSE TO THE REVIEWERS

NOTE TO THE REVIEWERS

We appreciate the helpful comments and constructive criticism offered by the reviewers. In order to fully address the reviews, we returned to the lab and conducted multiple additional in vitro and in vivo experiments, and this has led to 36 new figure panels. Our new data now include the observation of JHU083-mediated increased NO production in a macrophage model (**Fig. 6b**; this supports the metabolomic observation of elevated citrulline levels with JHU083) and the determination of DON levels in JHU083 treated lungs as being 0.856 nmol/g (**Fig. 2c**; a level 10,000x below the MIC of JHU083) a finding that supports JHU083 action is via a host-directed mechanism.

With these new data the revised manuscript is significantly improved as a result of the thoughtful comments made by all three reviewers. The point-to-point response to the reviewers, main manuscript, and supplementary information has been provided. We have also provided two versions of the main manuscript and supplemental information. One version has revised text highlighted in yellow, while the other is unhighlighted. Line numbers below refer to the yellow-marked version.

Reviewer #1 (Remarks to the Author):

1. The manuscript by Parveen et al. highlights the role of a known glutamine metabolism antagonist, JHU083 in inhibiting Mtb proliferation and growth in vitro and in vivo. The authors propose that JHU083-mediated glutamine metabolism inhibition results in dual antibacterial and host-directed activity against tuberculosis.

Major comments

Whereas the role of glutamine metabolism in T cell activation is described in the literature, the use of the glutamine metabolism inhibitor JHU083 resulting in T cell activation during Mtb infection is very interesting and relevant to the discovery and development of new HDTs for the control of TB. The study is well written, focused and the extensive flow cytometry data sets are appreciated. The biggest concerns include the use of selective representation of data sets from different animal models, and the lack of mechanism. For example, in case of the former, DON was used in the in vitro CFU assay instead of JHU083, BMDM from C57BL/6 mice were used whereas I29S2 mice were used for in vivo experiments, and C3HeB/FeJ mice were used to demonstrate histopathology. Also, different time points were used for CFU determination in I29S2 (2 and 5 week) and C3HeB/FeJ mice (4 weeks). These concerns could potentially be

addressed. Lastly, although inhibition of glutamine metabolism was not supported by the metabolomics data (although there appears to be a trend towards significance), and complicates the mechanism, my view is that this is not a critical caveat.

To address reviewer concerns regarding the mice strains and the time-points, we have repeated several experiments. The details of which are described below:

(1) Original **Fig. 1d** (*Mtb* growth inhibition in C57BL/6 BMDMs treated with DON). To address the inconsistent BMDM sources (C57BL/6 BMDMs while 129S2 mice were used elsewhere) and the omission of JHU083 in previous **Fig. 1d**, we have repeated this experiment using BMDMs isolated from 129S2 mice and with both DON and JHU083 treatment. This appears as new **Fig 1f**, and the old C57BL/6 data is now **Fig. S1a**. These new data also show that both DON and JHU083 inhibit *Mtb* growth in BMDM by 0.3 (DON) and 0.7 (JHU083) log₁₀ CFU units (**Fig 1f**). The text has been modified accordingly (**lines 129-131**).

(2) We originally tested the mouse vivo therapeutic efficacy of JHU083 in two separate mouse strains: 129S2 and C3HeB/FeJ mice strains. However, the reduction in lung bacillary burden was more pronounced in 129S2 mice (~1.9-2.5 log₁₀ CFU reduction, shown in **Fig. 2b**, $p < 0.05$) than in C3HeB/FeJ (~1.0 log₁₀CFU reduction, shown in **Fig. S3a**, $p < 0.05$). Also, there was a lower degree of intra-group variability for CFU counts in 129S2 mice. As a result of this observation, we used the 129S2 strain for most of the experiments described in this manuscript. We have added this explanation in the text (**lines 176-178**).

(3) The C3H/FeBJ mouse strain was used primarily to assess lung pathology in the context of JHU083 treatment as this is one of the very few mice strains known to form necrotic granulomas closely resembling the granulomas observed in human TB patients (Harper, Skerry et al. 2012, Ordonez, Tasneen et al. 2016, Urbanowski, Ordonez et al. 2020). We have added this explanation in the results section (**lines 159-162**). We have also repeated the histopathological analysis using *Mtb*-infected 129S2 mice lungs. We observed a 50% reduction in the lung lesion area, however, the data remained statistically insignificant ($P=0.07$, $n=3$ per group). We have included this data as new **Fig 2i** and **supplementary Fig S5** and have modified the text accordingly (**lines 170-174**).

(4) For in vivo CFU determination, the discrepancy in the time-points between 129S2 and C3HeB/FeJ mice arose because untreated C3HeB/FeJ mice, when infected with the same infection dose as 129S2 mice, consistently died between week 4 and 5 resulting in the early termination of the experiments. The C3HeB/FeJ CFU data shown in the **supplementary figure S3a** is ~4.5 weeks. We have modified the “Materials and Methods” section as well as the legend of **Fig 2** and **supplementary Figure S3a** to reflect the accurate time-points.

2. What is the rationale of using 129S2 mice that are typically used for transgenic/knockout model

development? Why were C3He or BALB/c strains not used? I see that for CFU assessment, C3HeB/FeJ mice were used in one panel but throughout the manuscript (flow data), 129S2 mice were used. The authors should provide an explanation.

Please see our answer to Reviewer 3's question #10 which was similar. Oh et al. 2019 demonstrated that JHU083 depletes myeloid derived suppressor cells (MDSCs) in the cancer microenvironment contributing to the strong anti-tumor effects of the drug. Based on this, we chose to test JHU083 activity in mouse strains with abundant MDSCs (eg, 129S2, C3HeB/FeJ) rather than mouse strains with relatively low levels of MDSCs (eg, Balb/c and C57BL/6) (Knaul, Jorg et al. 2014, Tsiganov, Verbina et al. 2014). Also, it is well-known that MDSCs facilitate tuberculosis pathogenesis and progression (du Plessis, Loebenberg et al. 2013, Dorhoi and Kaufmann 2015), and we reasoned that drug action may be easier to detect in mouse strains that are relatively more susceptible to *Mtb* disease progression. We have added this explanation in the Results sections (**Lines 142-145**) along with the appropriate references (**Reference numbers 7, 20 and 21**).

Based on Reviewer 1's recommendation, we also tested the efficacy of JHU083 in *Mtb*-infected C3H mice, and we found that JHU083 treatment did not reduce either the lung weight or the lung bacillary burden. This data is provided as a new **Supplementary Figure S3b**, and the text has been modified accordingly (**lines 159-164**).

3. In Figure S3 and S4, the gating strategy looks a bit subjective. What were the controls - FMO, isotype controls for generating the gates? Please include this information. Gates are shown without having a distinct zone and looks marginal. Can the authors comment on these issues perhaps in the figure legends?

While designing the flow panels, individual antibodies were titrated to identify the concentration with maximum specificity coupled with minimum possible spillover. The gating strategy was defined using single-stain and FMO controls (for low-expression markers). The figure legends of **Supplementary Fig S6, S7, S8, S12 and S13** have been modified to reflect this information.

4. Line 42; The authors mentioned reduced glutamine levels in the abstract; since these levels were not statistically significant different from the untreated group, I suggest they remove this statement. This will then be consistent with the Discussion.

As recommended, we have now removed this statement from the abstract.

5. I suggest the authors remove the last paragraph from the introduction as it is virtually identical to the abstract with a few extra details.

As recommended, we have removed the last paragraph of the introduction.

6. *Figure 1C; can the authors indicate in the legend the number of replicates, independent experiments, standard deviation? Given the significance of the study, the authors should have used a more rigorous approach for determining Mtb killing; e.g., CFU-based killing assays.*

Please see our answer to Reviewer 3's question #1 which was similar. As recommended, we have now determined the minimum bactericidal concentration (MBC), which is a CFU-based killing assay, of JHU083. As per this assay, JHU083 has an MBC value of 32 µg/ml, in addition to its MIC value of 1-2 µg/ml. Thus, JHU083 is bacteriostatic at the lower concentration and bactericidal at the higher concentration. We have included this data as new **Figure 1e** and have modified the text accordingly (**Lines 125-128**).

7. *Line 157; it will be helpful if the authors could provide a rationale for initiating treatment one day post infection.*

This is the first study testing the novel glutamine prodrug, JHU083, as a host-directed therapy for tuberculosis. Treatment was initiated early to give maximum chance to the drug to have an effect. Moving forward, we will be testing the therapeutic efficacy of the drug in both acute and chronic models of infection. We have modified the text to add this explanation (**Lines 478-483**).

8. *Line 179; SCID mice details are lacking in the Animal Infection Studies in the materials and methods section.*

We have included the information in the Materials and Methods section (**Line 73**).

9. *In lines 206 -209, how does increased naive T cells (CD4+ and CD8+ expressing CD44-CD62+) protect treated mice from TB compared to untreated mice?*

Wolf et al. *Nat Immunol* 2020 (ref. 47) have demonstrated that naïve T-cells despite maintaining a quiescent state can mount an effective and rapid immune response following activation (Wolf, Jin et al. 2020). Additionally, Orlando et al. (ref. 48) have identified a novel population of naïve CD4+ T-cell population that rapidly produce multiple cytokines in the blood of active TB patients (Orlando, La Manna et al. 2018). Recently, Daniel et al. have demonstrated that homing of naïve lymphocyte to the lung draining lymph nodes aid anti-TB T-cell response in mice (Daniel, Counoupas et al. 2023). These studies support the notion that naïve T-cells contribute to TB containment. We have added this additional information in the discussion (**Lines 380-388**). We have also added the explanation about follicular helper T-cells and the potential role of B-cell immune response in *Mtb* containment (**Lines 388-392**).

10. *Why are naive CD4 cells (Fig 4E) presented in % whereas naive CD8 cells (Fig 4G) are presented in*

MFI?

Thank you for pointing this out. To be consistent with the % Naïve CD4+ T-cell population (**Fig. 4E**), we have added **Fig. S9c** which shows the % Naïve CD8 T cells. The original **Fig. 4g** was gMFI of CD62L on CD8 cells which shows significant differences. Regarding the matter of showing percent population in certain panels and gMFI in other panels, we have now added text to explain that we present gMFI for low abundance cell surface markers and transcription factors (**line 840 and lines 855-6**).

11. Line 267; given that the reduction in glutamine levels is not statistically significant I suggest the authors change “The most notable changes...” part of the sentence to indicate this fact. I suggest authors concisely state why changes in glutamine levels were not observed.

Please see our response to Reviewer 2, question 4 and Reviewer 3, question 13 which were on a similar point. We have changed the text to make it clear that the Gln levels showed no statistically significant differences between control and JHU083-treated groups. We have also repeated the mouse lung metabolite experiment to quantify glutamine and glutamate levels in the whole lung tissue using targeted metabolomics. We did not detect any consistent significant changes in these values in the PBS and JHU083-treated groups. This new information has been added as the **supplementary Figure S20** and modified text (**Lines 296-298**). We have also included additional text to highlight the fact that the glutamine levels changes were not observed and have listed potential reasons for this in the discussion (**Lines 441-458**):

“Interestingly, unlike the cancer studies(Leone, Zhao et al. 2019, Oh, Sun et al. 2020), we did not observe a statistically significant difference in glutamine levels between control and JHU083-treated lungs. One reason for this may be that the cancer studies measured glutamine levels in heterotopic flank tumors while we evaluated whole lung. Indeed, JHU083 was originally selected for further study because it was found to be preferentially metabolized from pro-drug to DON in the tumor microenvironment (Rais, Jančařík et al. 2016, Lemberg, Vornov et al. 2018). Indeed, following JHU083 treatment we found rather low levels of DON in lungs (0.856 nmol/g) while its Cmax in murine MC38 tumors (5.38 nmol/g) and murine plasma (4.1 nmol/ml) is considerably higher (Leone, Zhao et al. 2019). Thus, it is possible that the JHU083-mediated immune cell shifts we observed in lung reflect the action of JHU083 on lymphoid and myeloid cells in other compartments such as blood, spleen or bone marrow with subsequent migration of cells to the lung and that Gln levels in the lung itself do not play a causal role. Another possibility is that JHU083 affects Gln levels in particular lung cell subsets which we could not evaluate by measuring Gln in the whole lung homogenate MeOH extract. It should also be noted that there is heavy reliance of lung tissue on the de novo Gln synthesis from glutamate and ammonia (Pan, Wasa et al. 1995, Labow, Abcouwer et al. 1998), and rodent lungs have also been shown to accumulate as much Gln as skeletal muscle (the tissue with the highest free Gln concentration (Souba, Herskowitz et al. 1990)). These factors indicating high Gln pools in the lung, along with the relatively low level of DON we observed, support the concept that JHU083-mediated lung immune cell reprogramming may occur outside of the lung.”

12. Line 337; remove the Unpublished Data.

As recommended, we have now removed the reference to unpublished data.

13. In Fig 4 and Fig 5, the immunology data are inconsistent as some of the cell markers are shown as % (4A, E, F and Fig 5A, B, E, F, G), whereas the other panels were presented in MFI. Why is this the case?

We specifically chose gMFI for the markers that are low abundance cell surface markers (example: CD206) and transcription factors (example: BCL6). We have modified the figure legends of **Fig 4, 5, S10, S14** to reflect this information.

Minor comments:

14. Line 167 and line 168 – Fig 2C should be Fig 2D and Fig 2D should be Fig 2C respectively.

Thank you for pointing this out. We have made the correction.

15. In figure 2E, the lung weight should be in terms of mg, or change the axis accordingly. Weight of lungs cannot be 200g.

Thank you. We have corrected the y-axis label.

16. In the method section of the supplementary file, the method for “Lung histopathology estimation” (in line 71) indicate 5 weeks of post-infection, whereas in the result section of the manuscript (line 169-170) it indicates 4 weeks of infection/treatment was used for histopathological examination.

The accurate time-point was 4.5 weeks. We have addressed the discrepancy and now have a consistent timeline across the various sections of the manuscript.

17. Fig 3C indicates the SCID mouse body weight was measured for up to 7 wk (i.e., 49 days). However, survival data indicate that both PBS and JHU083-treated mice die before 7 wk time. Is this a mistake in numbering weeks post- infection, instead of 6 it is written as 7 weeks (as each time point is showing 1 wk difference instead of the last time point).

We have modified the label to rectify the mistake.

18. Line 214 and line 218 - Fig 3I & 3J should be Fig 4I & 4J and Fig 3K & 3L should be Fig 4K & 4L.

We have now corrected the mistake

19. Line 264 and line 372 – there is no “Supplementary Table S1”.

We have provided the information as a separate excel sheet titled “Supplementary Table S1”.

Reviewer #2 (Remarks to the Author):

The work entitled “Glutamine metabolism inhibition has dual immunomodulatory and antibacterial activities against Mycobacterium tuberculosis” by Parveen and others details studies testing whether a glutamine metabolism antagonist colloquially named ‘JHU083’, an orally available prodrug of the 6-diazo-5-oxo-L-norleucine, would further reduce immunopathology associated with tuberculosis infection. Administration of JHU083 was tested using aerosol infection in I29S2 mice and rifampin was used as a comparator along approximately the same timeline. Numerous biological endpoints were measured to assess performance of JHU083, and this included bacillary loading in the lung, cell subset- specific activity, inflammatory mediators, global metabolomics (of the whole lung), and pathology spec. to granuloma formation.

There are no major concerns with this work that would completely halt recommendation for publication. However, there are some questions about approach in the general experimental design that are questionable and that must be addressed by the authors before proceeding.

1. There was little to no time spent on understanding (or commenting on) basic pharmacokinetics of this prodrug. It was clearly stated that there was transformation that took place once in vivo (to 6-diazo-5-oxo-L-norleucine) and the corresponding oral bioavailability, which should have ‘fueled the fire’ to perform a careful pharmacokinetics study prior to a therapeutics evaluation. Yet, studies with 6-diazo-5-oxo-L-norleucine which showed minimal benefit and GI-related ‘difficulties’ was the only mention of anything remotely resembling an idea about pharmacokinetics. What about the esterases that transformed the prodrug? Was this measured in the context of dosing, and how variable is the in vivo concentrations in the context of oral dosing? Is there any difference in another route of administration that may effect transformation?

A closely related drug to JHU083, namely DRP-104 is in the phase I/II trials for solid tumors. Both the pharmacokinetics and toxicity profile of JHU083 in animal models has been studied and published, and we appreciate Reviewer 2’s comments that this information needs to be included in our paper. Some of the studies describing the pharmacokinetics of JHU083 using various route of administration in preclinical models are: Zhu, Xiaolei, et al. Neuropsychopharmacology 44.4(2019): 683-694 (**ref. 26**); Nedelcovych, Michael T., et al. Journal of Neuroimmune Pharmacology 14 (2019):391-400 (**ref. 31**), and we now cite these papers. In addition, we also performed additional pharmacologic studies to quantify the level of the mature drug, DON, in *Mtb* infected lungs. Based on this LC/MS based assay, we found that 30-40 min after receiving a 1

mg/kg dose of JHU083, the DON level in JHU083-treated, *Mtb* infected lungs were 0.896 nMoles/g. This level is comparable to the Cmax levels in mouse brain (0.85 nmol/g) and lower than its mouse Cmax in plasma (4.1 nmol/g)). The data has been added as **Fig. 2c**, text has been modified accordingly (**Lines 156-159, 322-324, 467-469**) and references have been included (**References 22-26, 31**)

2. Only one dose (1 mg/kg) of JHU083 was presented in this work. Was other dosing regiments attempted, and if not, then why would only one dose regiment be used when considering evaluation of any treatment. Also, only one regiment (daily) was used in the evaluation. With virtually no pharmacokinetics data to inform dose or regiment, how was the decision made to arrive at these parameters? Either an incredibly risky or lucky way to plan this study.

We thank Reviewer 2 for this comment, as the previous manuscript should have contained information on dose selection. We selected the 1 mg/kg dose (employed in this study) because this dose has shown efficacy and tolerability in several other murine models (eg. The two articles cited above (**ref. 26, 31**) as well as Hollinger et al. *Neurol Neuroimmunol Neuroinflamm* Nov 2019, 6 (6) e609 (**ref. 23**); Riggle et al. *Proc Natl Acad Sci USA* 115.51 (2018): E12024-E12033 (**ref. 22**). Hanaford, Allison R., et al. *Transl Oncol* 12.10 (2019):1314-1322 (**ref. 25**), and Hollinger, Kristen R., *J Alzheimer's Dis* 77.1 (2020): 437-447) (**ref. 24**). We also tested two different dosing regimens: (1) Daily regimen: 1 mg/kg dose per day for first week, followed by 0.3 mg/kg daily (5/7, M-F) and (2) Alternate regimen: 1 mg/kg dose per day for first week, followed by 1 mg/kg on alternate days (3/7, Mon, Wed and Fri). Both regimens led to similar reduction in the lung bacillary burden and lung weight in murine model of tuberculosis. We have provided this information as a new **Supplementary Figure S2**. The text has been modified accordingly (**Lines 152-156**), and new references have been included (**References 22-26, 31**). Figure legends have been modified to state precisely which regimen was used.

3. This work is a comprehensive and at times overwhelming assessment of the cellular mechanism-based changes associated with introduction of an experimental reagent drug in a murine model of tuberculosis. The amount of endpoint analysis in the context of measured endpoints is borderline unfocused – there is so much endpoint data provided with minimal planning on the basic parameters necessary to determine optimal administration and route relative to kinetics of JHU083 (referral to earlier comments). Accordingly, there is a concern with organization of the data as it relates to assessment. It seems that there is little regard for omission of some data with no explanation, wherein it (seems) to the reader is to overwhelm with data, rather than provide a careful interpretation of why one or the other measurement was excluded. A clear example is the exclusion of rifampin comparator in Figure 2C, 2D, and 2G.; body wt gain, time to death, and % granulomas, three otherwise major indicators that were excluded from graphs, yet in

Fig. 2E lung weight rifampin appears again. Similarly, in Fig 3 that details results with the SCID mice, rifampin appears as a comparator. The immediate concern is why this data for the comparator was excluded in Fig. 2, but more generally is this thematic for understandably densely packed report.

We appreciate the reviewer 2's point regarding the exclusion of the rifampin comparator. We have now included the information on the rifampicin comparator in the manuscript for all of the panels in Fig. 2 except Fig. 2d (time-to-death). Unfortunately, separate time-to-death studies were performed for JHU083 vs PBS and for RIF vs. PBS, and because time-to-death is highly dependent on the precise inoculum we feel that the two time-to-death studies cannot be directly compared on the same graph. The RIF vs PBS time-to-death data are provided in the new **Supplementary Figure S4c** and the text has been modified accordingly (**Lines 174-176**).

4. A similar and related concern is the inclusion of analysis that (an opinion) adds minimally to the overall support of the conclusions of this work. A clear example is inclusion of the metabolomics data, that is already discounted by the authorship because of whole lung digestion rather than specific cell subsets, and largely uninterpretable heatmap presentation with relatively weak statements in the legend of the figure (actually no conclusions) and in the text, line 267, "The most notable changes in JHU083-treated lungs were as follows: (i) A ~17% reduction in the glutamine levels ($P = 0.17$; Fig. 6A)". The P value provided is nonsignificant, but it is "the most notable change"? It is hard to understand why this would be included in this manuscript with so many other endpoints that show benefit of JHU083 administration.

Please see our response to Reviewer 1, question 11 and Reviewer 3, question 13, which were on a similar point. We agree with the reviewer. We have made the observations in the whole lung using a high-throughput untargeted metabolomics. As pointed out by reviewer 2, the lower concentration of glutamine was not statistically significant. We have now repeated the assay again using targeted metabolomics assay. Unlike cancer studies, we found no statistically significant difference between control and JHU083-treated lungs. This new information has been added as the **supplementary Figure S20** and modified text (**Lines 296-298**). We have also included additional text to highlight the fact that the glutamine levels changes were not observed and have listed potential reasons for this in the discussion (**Lines 441-458**):

"Interestingly, unlike the cancer studies(Leone, Zhao et al. 2019, Oh, Sun et al. 2020), we did not observe a statistically significant difference in glutamine levels between control and JHU083-treated lungs. One reason for this may be that the cancer studies measured glutamine levels in heterotopic flank tumors while we evaluated whole lung. Indeed, JHU083 was originally selected for further study because it was found to be preferentially metabolized from pro-drug to DON in the tumor microenvironment (Rais, Jančařík et al. 2016, Lemberg, Vornov et al. 2018). Indeed, following JHU083 treatment we found rather low levels of DON in lungs (0.856 nmol/g) while its Cmax in murine

MC38 tumors (5.38 nmol/g) and murine plasma (4.1 nmol/ml) is considerably higher (Leone, Zhao et al. 2019). Thus, it is possible that the JHU083-mediated immune cell shifts we observed in lung reflect the action of JHU083 on lymphoid and myeloid cells in other compartments such as blood, spleen or bone marrow with subsequent migration of cells to the lung and that Gln levels in the lung itself do not play a causal role. Another possibility is that JHU083 affects Gln levels in particular lung cell subsets which we could not evaluate by measuring Gln in the whole lung homogenate MeOH extract. It should also be noted that there is heavy reliance of lung tissue on the de novo Gln synthesis from glutamate and ammonia (Pan, Wasa et al. 1995, Labow, Abcouwer et al. 1998), and rodent lungs have also been shown to accumulate as much Gln as skeletal muscle (the tissue with the highest free Gln concentration (Souba, Herskowitz et al. 1990)). These factors indicating high Gln pools in the lung, along with the relatively low level of DON we observed, support the concept that JHU083-mediated lung immune cell reprogramming may occur outside of the lung.”

Very minor

5. Line 88, ‘track’ should be ‘tract’

Thank you. We have made the correction.

6. Line 336-337, please consider removing any references to ‘unpublished data’ as this is completely unhelpful Line 374,

As recommended, we have removed reference to unpublished data (**Lines 366-368**)

7. 5-HIAA is undefined

Thank you. We have now defined 5-HIAA as 5-HydroxyIndole acetic acid (**Line 290-291**).

8. Fig S2, legend, consider changing ‘two-times’ to ‘twice’

Thank you. We have implemented the suggestion.

9. Fig S11, Fig S12, the graphic quality of these two heatmaps are horrible

We have now increased the font size to improve the quality of the heatmaps. These heatmaps are now designated as **supplementary Figure S16 and S17**.

Reviewer #3 (Remarks to the Author):

Authors within this group have developed a pro-drug for DON which has demonstrated remarkable potency in treating a variety of tumors in mice. These authors have demonstrated that JHU083 affects tumor-infiltrating T cells by increasing expression of markers of activation and proliferation and by decreasing the

number and activity of myeloid-derived suppressor cells. In this work, the authors explore the effect of JHU083 on a murine model of tuberculosis. Firstly, based on the known role of Mtb glutamine synthetase in pathogenesis, the authors show that the compound has a direct growth inhibitory effect on the bacterium, but the mechanism is not explored even though this would be easy to do.

Next the authors show that treatment of infected mice has an extremely modest effect on bacterial burdens in organs of infected mice although by treating the mice with a sub-optimal dose of Rifampicin, the result seem good. The compound does, however, have a dramatic effect on survival, body weight and lung weight of infected mice suggesting that something important is happening. A non-standard mouse strain is used which makes data interpretation a bit complex and the lack of efficacy in SCID mice is promising although the commonly used immunocompetent control strain is not compared which makes benchmarking tough. The authors next analyze different cell types in the lung but although statistically measurably differences are seen, there is no clear answer as to the functionally relevant cell population induced by JHU083. In many cases, similar changes are seen for Rif-treated animals. The effect on myeloid derived suppressor cells is suggestive but the more convincing follow-up work reported previously by some of the authors in this group is not done. In contrast to the cancer work where the tumor microenvironment is often hypoxic, the authors do not test the compound in a necrotic granuloma model in mice even though the mouse strain that can develop such granulomas given the right experimental setup is used. Thus, comparing effects in tumor experiments reported in the literature with the effects seen here is impossible. There are differences but some of these are also seen in Rifampicin- treated mice. Metabolites are measured and the most important metabolite that could possibly explain the specific effect of JHU083 is quinolinic acid, but the authors stop short of demonstrating that IDO activity is key to the host- directed effect of JHU083.

In summary, the results are interesting and point to a host-directed benefit of JHU083 in chemotherapy of Mtb-infected mice. A few key experiments would allow the authors to address some of their hypotheses more conclusively.

Specific comments

1. DON and JHU083 inhibit MTb growth in vitro. If the assumption is that these inhibit the essential GlnAI enzyme, gln supplementation would rescue (as reported for MSO). Does gln or glu supplementation affect the MIC? Is JHU083 static or cidal against Mtb in vitro?

We thank Reviewer 3 for this helpful suggestion. Earlier studies suggest that DON has much more complex mechanism of action than MSO. DON has been reported to selectively block numerous glutamine-utilizing enzymes including carbamoyl phosphate synthase, cytidine triphosphate synthase, phosphoribosyl formylglycinamide synthetase (PFAS), guanosine monophosphate synthetase, phosphoribosyl pyrophosphate aminotransferase, nicotinamide adenine dinucleotide synthase, asparagine synthase, glutaminase and glutamine synthetases (Lemberg, Vornov et al. 2018) (**ref. 12**).

Nevertheless, we agree that testing for Gln-interference with the JHU083 MIC is a good idea. We performed the Alamar blue assay with JHU083 in presence of increasing concentrations of glutamine from 3.2 – 32 μ M (ie, or 0.5x to 5x the JHU083 MIC) and found no change in the JHU083 antibacterial activity (MIC = 2 μ g/ml at all levels of Gln). We have added these data as **Main Figure 1d** and modified the text accordingly (**Lines 123-25**).

Another line of evidence comes from the BMDM experiment as despite the presence of glutamine in the cell culture media (in millimolar concentration range), both DON and JHU083 reduced the bacillary burden (**Figure 1f**). We have added this explanation in the discussion section too (**Lines 358-366**).

Regarding the query of whether JHU083 is bacteriostatic or bactericidal. We measured the MBC to be 32 μ g/ml, while the MIC is 1-2 μ g/ml. Thus, JHU083 is bacteriostatic at the lower concentration and bactericidal at the higher concentration. We have included this data as new **Figure 1e** and have modified the text accordingly (**Lines 125-128**). Please see our answer to Reviewer 1's question #6.

2. Is JHU083 converted to DON by Mtb?

Mycobacterium tuberculosis expresses several proteases and esterases that are either secreted or cell-wall associated. It is possible that bacterial enzymes convert JHU083 into DON, but it is also clear from the cancer work on JHU083 that host proteases and esterases also mediate the conversion. The direct antibacterial activity of JHU083 in vitro (**Fig 1c** and **1e**) and the fact that JHU083 and DON have the same MIC of 1-2 μ g/ml strongly suggests that bacterial enzymes mediate the conversion, however, additional experiments will be required to conclusively prove this.

3. The antibacterial efficacy against Mtb growing in macrophages suggests a rather limited effect against Mtb in host cells but as mentioned by the authors, could reflect poor access to the phagosome. Alternatively, the Gln in the tissue culture medium could rescue inhibition.

We have performed Alamar blue assay with JHU083 in presence of increasing concentration of glutamine. However, glutamine supplementation as high as 32 μ M (5x the JHU083 MIC) did not hamper the antibacterial activity of JHU083 (MIC = 2 μ g/ml). We have added this data in **Main Figure 1d** and modified the text accordingly (**Lines 123-25**). While it is possible that millimolar concentration of glutamine in RPMI glutamax media (2-6 mM Gln) used for the macrophage assay interferes with the activity of the drug. However, glutamine is the most abundant amino acid present in the lung, and JHU083 still remained effective against *Mtb*. We have added this explanation in the discussion section too (**Lines 357-366**).

4. Fig. 2: mice are treated 1 day after infection which is an extremely early stage of infection. Did the authors attempt treatment at later stages?

This is the first study testing JHU083, a novel glutamine prodrug, as a host-directed therapy for tuberculosis. Treatment was initiated early to give maximum chance to the drug to have a therapeutic effect. Moving forward, we will be testing the therapeutic efficacy of the drug in both acute and chronic models of infection. We have modified the text to add the explanation (**Lines 478-483**).

5. Fig. 2: Rif dosing at 1.25mg/kg seems extremely suboptimal. Why did the authors choose this very low dose?

We thank the reviewer for calling this error to our attention. The actual rifampicin dosing utilized in the study was 12.5 mg/kg (slightly higher than the standard dose of 10 mg/kg). We have now rectified the mistake and have made changes accordingly.

6. The dosing strategy for JHU083 is similar to that used in Leone et al 2019 and Oh et al. 2019 – for the first 5 days mice are given 1mg/kg but the dose is 3-fold lower later. Is this due to toxicity?

We used two different dosing regimens, (1) Daily regimen: 1 mg/kg dose per day for first week, followed by 0.3 mg/kg daily and (2) Alternate regimen: 1 mg/kg dose per day for first week, followed by 1 mg/kg on alternate days (Mon, Wed and Fri). Both regimens led to similar reduction in the lung bacillary burden and lung weight in murine model of tuberculosis. We have provided this information as a new **Supplementary Figure S2**. The text has been modified accordingly (**Lines 152-156**). The toxicity profile of JHU083 has been extensively tested in various preclinical models, and the two dosing regimens used in this work are those optimized for tolerability and efficacy (**References 22-25**).

7. The C3HeB/FeJ mouse model can form necrotic granulomas. However, the necrotic granulomas only develop after a month, usually at lower implantation doses. Thus, stating that the activity of JHU083 was assessed in a mouse model which develops necrotic granulomas is misleading to most readers.

We agree with the reviewer that in C3HeB/FeJ mice necrotic granulomas usually appear 6 weeks post-infection. However, while this is indeed the case with low-dose challenge (100 CFU or less), necrosis occurs sooner with high burden *Mtb* challenge. In our C3HeB/FeJ work we challenged with 200-300 CFU. As may be seen in **Fig. 1f**, the lesions in the PBS-treated mice at 4.5 weeks post-infection do in fact show necrosis. In light of the reviewer's helpful comment, we have modified the text to reflect that the histology was performed in mouse model of necrotic granuloma, however, at the time of pathology, only early necrotic granulomas were present (**Lines 89-92 of Supplemental Methods**).

8. Fig. 2G: granuloma numbers were measured – what did the authors define as a granuloma? This is not mentioned in the M&M.

We have defined granulomas as an aggregation of epithelioid macrophages which may also contain giant cells and may or may not be surrounded by a cuff of lymphocytes. We have now added this definition to the Materials and Methods section (**Lines 93-95 of Supplemental Methods**).

9. Lines 167 and 168: reference to the MTD and body weight panel is switched (Fig. 2C versus 2D)

We have now rectified this mistake.

10. Fig. 3: It's frustrating that the authors don't show the efficacy results for JHU083 in immunocompetent Balb/c mice. The authors routinely use Balb/c mice for published work thus it seems strange that JHU083 efficacy in this mouse strain is not reported. This would be important to benchmark these results.

Please see our answer to Reviewer 1's question #2 which was similar. We have not tested Balb/c and have instead chosen 129S2 and C3HeB/FeJ mice considering that JHU083 has been shown to reduce MDSCs in cancer models (Oh et al. 2020; ref. 7) and 129S2 and C3HeB/FeJ mice are known to demonstrate higher frequencies of MDSCs upon *Mtb* infection. However, we have now repeated the infection study in C3H strain that is the parent strain of C3HeB/FeJ and have shown that JHU083 does not reduce either lung bacillary burden or lung weight. This data has been added to **Supplementary Figure S3b** and the text has been modified accordingly (**Lines 159-164**). The explanation for the strain choice has also been added (**Lines 142-145**).

11. In all cell analyses, the uninfected mouse control group treated with JHU083 is not shown. This is an important control for this mouse strain.

As recommended, we have now tested the effect of JHU083 in uninfected 129S2 mice and have shown that JHU083 treatment does not affect the frequencies of various T-cell subsets in uninfected mice lungs. However, changes in the frequency of B-cells, monocytic MDSCs and interstitial macrophages were observed. The data has been added as two new **Supplementary Figures S11 and S15**. The text has been modified accordingly (**Lines 223-225 & 262-264**).

12. It is intriguing that the T cells that infiltrate the lungs of JHU083 Mtb infected mice express high levels of markers for naïve T cells and follicular T cells. This response is transient. Thus, it is unclear how this increase in T cells results in the small differences in organ bacterial burdens and dramatic changes in mortality. Fig. 4: Despite many changes being statistically significant, the actual changes are often quite

small (eg. in untreated mice 4% of CD45+ cells are naïve CD4+ cells whereas this increases to maybe 7% in JHU083 treated mice) which further raises questions about which of the observed changes are important for the apparent positive effect of JHU083. Similarly, the monocytic myeloid-derived suppressor cells are statistically different but the % change is small (2% versus 3% at week 2 for untreated vs JHU083- treated mice, respectively). Many of the cellular differences are comparable to that measured in the Rif-treated group. If JHU083 is having specific immunomodulatory effects, one would expect the relevant functionally important differences to be specific for JHU083.

We agree with the reviewer that JHU083 as a monotherapy leads to transient changes in immune cell frequencies in the *Mtb*-infected lungs compared to the control. However, these transient changes were sufficient to decrease 99% of the initial lung bacillary burden (~1.9-2.5 log₁₀ reduction). We anticipate that combining JHU083 with standard drug regimens may lead to even greater reductions in the lung bacillary burden, and such combination treatment regimens will be an important focus in the continuation of this work.

We also agree that about half of immune cell changes we observed by flow cytometry occurred in both the JHU083- and RIF-treated groups. However, another half of the changes—particularly those with T cells--were specific for JHU083 treatment. These differential results point toward the conundrum that we have faced while attempting to parse out the mechanism of action of JHU08. While it is true that lowered bacillary burden due to direct antibacterial activity may lead to changes in the immune cell populations, JHU083's inability to reduce bacillary burden in immunocompromised mice strongly suggests that JHU083's immunomodulatory activity is prominent (**Fig 3**). As expected, RIF treatment reduced bacillary burden in both immunocompetent and immunocompromised mice infected with *Mtb* (**Fig 3**). Additionally, as per the results of the additional pharmacokinetic study that was performed to address reviewers' concerns, the DON levels in *Mtb*-infected lungs tissues are at least 10,000-fold lower than the MIC value of JHU083 (**Fig 1c**) which further supports a host-directed mechanism. Accordingly, we have modified the text in the Discussion section of the manuscript to provide a potential explanation (**Lines 467-469**).

13. The authors observe lower levels of gln. Previous work had demonstrated increased gln/glu ratios which seemed to make sense when considering the mechanism of JHU083. What were the Gln/glu ratios?

Please see our response to Reviewer 1, question 11 and Reviewer 2, question 4, which were on a similar point. We have made the observations in the whole lung using a high-throughput untargeted metabolomics. As pointed out by Reviewer 2, the lower concentration of glutamine was not statistically significant. We have now repeated the whole lung metabolite studies using a targeted metabolomics assay. We found no difference in either the concentration of glutamine or the glutamine:glutamate ratio between JHU083-treated and untreated controls. This new information has been added as the **supplementary Figure S20** and

modified text (**Lines 296-298**). We have also included additional text to highlight the fact that the glutamine levels changes were not observed and have listed potential reasons for this in the discussion (**Lines 441-458**):

“Interestingly, unlike the cancer studies(Leone, Zhao et al. 2019, Oh, Sun et al. 2020), we did not observe a statistically significant difference in glutamine levels between control and JHU083-treated lungs. One reason for this may be that the cancer studies measured glutamine levels in heterotopic flank tumors while we evaluated whole lung. Indeed, JHU083 was originally selected for further study because it was found to be preferentially metabolized from pro-drug to DON in the tumor microenvironment (Rais, Jančařík et al. 2016, Lemberg, Vornov et al. 2018). Indeed, following JHU083 treatment we found rather low levels of DON in lungs (0.856 nmol/g) while its Cmax in murine MC38 tumors (5.38 nmol/g) and murine plasma (4.1 nmol/ml) is considerably higher (Leone, Zhao et al. 2019). Thus, it is possible that the JHU083-mediated immune cell shifts we observed in lung reflect the action of JHU083 on lymphoid and myeloid cells in other compartments such as blood, spleen or bone marrow with subsequent migration of cells to the lung and that Gln levels in the lung itself do not play a causal role. Another possibility is that JHU083 affects Gln levels in particular lung cell subsets which we could not evaluate by measuring Gln in the whole lung homogenate MeOH extract. It should also be noted that there is heavy reliance of lung tissue on the de novo Gln synthesis from glutamate and ammonia (Pan, Wasa et al. 1995, Labow, Abcouwer et al. 1998), and rodent lungs have also been shown to accumulate as much Gln as skeletal muscle (the tissue with the highest free Gln concentration (Souba, Herskowitz et al. 1990)). These factors indicating high Gln pools in the lung, along with the relatively low level of DON we observed, support the concept that JHU083-mediated lung immune cell reprogramming may occur outside of the lung.”

14. There is a small increase in citrulline, and the authors hypothesize that this could be due to iNOS induction and activity. There are inhibitors available to directly test this although not recommended by this reviewer since Rif treatment has a similar effect. The increased citrulline levels contrasts with the decreases in tumor citrulline levels reported in JHU083 treated mice (Oh et al. 2019). Oh et al. (2019) reported decreased kynurenine levels in lung tumors of JHU083-treated mice. In this work only quinolinic acid, a potential byproduct of kynurenine metabolism, is detected. Was kynurenine not detected?The model if Fig. 7 suggests that JHU083 is responsible for disease regression by virtue of the lower production of kynurenine and increased production of nitric oxide. The problem with the NO-mediated control aspect of this model is that Rif-treated mice had similar enhanced nitric oxide levels. However, the effect of IDO is rather appealing since IDO activity is a known suppressor of T cell function and enhances the activity of myeloid-derived suppressor cells. It would have been wonderful to see the authors validate this prediction using an IDO inhibitor. Note that IDO inhibitors have been used for host-directed therapy studies for animal models of TB.

We thank Reviewer 3 for these excellent suggestions. We followed up on the concept that the higher citrulline level in JHU083-treatment group may have been related to higher NO production. Using BMDMs from 129S2 mice and the Griess-reagent based NO assay we observed that JHU083-treated macrophage exhibited significantly higher NO concentrations compared to PBS or Isoniazid treatment. This analysis clearly demonstrated that JHU083 treatment leads to the induction of NO in macrophages potentially via iNOS. This data is shown as **Figure 6 panel b**. We have modified the text accordingly (**Lines 42-44, 281-288, 401-402 & 491**). We have also modified **Fig 7** to reflect the production of NO.

Also, on the recommendation of Reviewer 3, we tested the levels of IDO1 enzyme immunoreactivity by Western blot in the whole lung lysate from Mtb-infected mice receiving PBS, JHU083, and rifampicin treatment, but we did not find a statistically significant differences in the IDO1 levels between PBS- and JHU083-treated lungs. These data have been added as **Supplementary Figure S19** and text has been added (**Lines 293-296**).

REFERENCES:

- Daniel, L., C. Counoupas, N. D. Bhattacharyya, J. A. Triccas, W. J. Britton and C. G. Feng (2023). "L-selectin-dependent and -independent homing of naïve lymphocytes through the lung draining lymph node support T cell response to pulmonary Mycobacterium tuberculosis infection." *PLoS Pathog* **19**(7): e1011460.
- Dorhoi, A. and S. H. Kaufmann (2015). "Versatile myeloid cell subsets contribute to tuberculosis-associated inflammation." *Eur J Immunol* **45**(8): 2191-2202.
- du Plessis, N., L. Loebenberg, M. Kriel, F. von Groote-Bidlingmaier, E. Ribechini, A. G. Loxton, P. D. van Helden, M. B. Lutz and G. Walzl (2013). "Increased frequency of myeloid-derived suppressor cells during active tuberculosis and after recent mycobacterium tuberculosis infection suppresses T-cell function." *Am J Respir Crit Care Med* **188**(6): 724-732.
- Harper, J., C. Skerry, S. L. Davis, R. Tasneen, M. Weir, I. Kramnik, W. R. Bishai, M. G. Pomper, E. L. Nuernberger and S. K. Jain (2012). "Mouse model of necrotic tuberculosis granulomas develops hypoxic lesions." *J Infect Dis* **205**(4): 595-602.
- Knaul, J. K., S. Jorg, D. Oberbeck-Mueller, E. Heinemann, L. Scheuermann, V. Brinkmann, H. J. Mollenkopf, V. Yermeev, S. H. Kaufmann and A. Dorhoi (2014). "Lung-residing myeloid-derived suppressors display dual functionality in murine pulmonary tuberculosis." *Am J Respir Crit Care Med* **190**(9): 1053-1066.
- Labow, B. I., S. F. Abcouwer, C. M. Lin and W. W. Souba (1998). "Glutamine synthetase expression in rat lung is regulated by protein stability." *Am J Physiol* **275**(5): L877-886.
- Lemberg, K. M., J. J. Vornov, R. Rais and B. S. Slusher (2018). "We're Not "DON" Yet: Optimal Dosing and Prodrug Delivery of 6-Diazo-5-oxo-L-norleucine." *Mol Cancer Ther* **17**(9): 1824-1832.
- Leone, R. D., L. Zhao, J. M. Englert, I. M. Sun, M. H. Oh, I. H. Sun, M. L. Arwood, I. A. Bettencourt, C. H. Patel, J. Wen,

A. Tam, R. L. Blosser, E. Prchalova, J. Alt, R. Rais, B. S. Slusher and J. D. Powell (2019). "Glutamine blockade induces divergent metabolic programs to overcome tumor immune evasion." Science **366**(6468): 1013-1021.

Oh, M. H., I. H. Sun, L. Zhao, R. D. Leone, I. M. Sun, W. Xu, S. L. Collins, A. J. Tam, R. L. Blosser, C. H. Patel, J. M. Englert, M. L. Arwood, J. Wen, Y. Chan-Li, L. Tenora, P. Majer, R. Rais, B. S. Slusher, M. R. Horton and J. D. Powell (2020). "Targeting glutamine metabolism enhances tumor-specific immunity by modulating suppressive myeloid cells." J Clin Invest **130**(7): 3865-3884.

Ordonez, A. A., R. Tasneen, S. Pokkali, Z. Xu, P. J. Converse, M. H. Klunk, D. J. Mollura, E. L. Nuermberger and S. K. Jain (2016). "Mouse model of pulmonary cavitary tuberculosis and expression of matrix metalloproteinase-9." Dis Model Mech **9**(7): 779-788.

Orlando, V., M. P. La Manna, D. Goletti, F. Palmieri, E. Lo Presti, S. A. Joosten, C. La Mendola, S. Buccheri, T. H. M. Ottenhoff, F. Dieli and N. Caccamo (2018). "Human CD4 T-Cells With a Naive Phenotype Produce Multiple Cytokines During Mycobacterium Tuberculosis Infection and Correlate With Active Disease." Front Immunol **9**: 1119.

Pan, M., M. Wasa, U. Ryan and W. Souba (1995). "Inhibition of pulmonary microvascular endothelial glutamine transport by glucocorticoids and endotoxin." JPEN J Parenter Enteral Nutr **19**(6): 477-481.

Rais, R., A. Jančařík, L. Tenora, M. Nedelcovych, J. Alt, J. Englert, C. Rojas, A. Le, A. Elgogary, J. Tan, L. Monincová, K. Pate, R. Adams, D. Ferraris, J. Powell, P. Majer and B. S. Slusher (2016). "Discovery of 6-Diazo-5-oxo-l-norleucine (DON) Prodrugs with Enhanced CSF Delivery in Monkeys: A Potential Treatment for Glioblastoma." J Med Chem **59**(18): 8621-8633.

Souba, W. W., K. Herskowitz and D. A. Plumley (1990). "Lung glutamine metabolism." JPEN J Parenter Enteral Nutr **14**(4 Suppl): 68s-70s.

Tsiganov, E. N., E. M. Verbina, T. V. Radaeva, V. V. Sosunov, G. A. Kosmiadi, I. Y. Nikitina and I. V. Lyadova (2014). "Gr-1dimCD11b+ immature myeloid-derived suppressor cells but not neutrophils are markers of lethal tuberculosis infection in mice." J Immunol **192**(10): 4718-4727.

Urbanowski, M. E., A. A. Ordonez, C. A. Ruiz-Bedoya, S. K. Jain and W. R. Bishai (2020). "Cavitary tuberculosis: the gateway of disease transmission." Lancet Infect Dis **20**(6): e117-e128.

Wolf, T., W. Jin, G. Zoppi, I. A. Vogel, M. Akhmedov, C. K. E. Bleck, T. Beltraminelli, J. C. Rieckmann, N. J. Ramirez, M. Benevento, S. Notarbartolo, D. Bumann, F. Meissner, B. Grimbacher, M. Mann, A. Lanzavecchia, F. Sallusto, I. Kwee and R. Geiger (2020). "Dynamics in protein translation sustaining T cell preparedness." Nat Immunol **21**(8): 927-937.

REVIEWERS' COMMENTS

Reviewer #1 (Remarks to the Author):

The concerns I had raised have been adequately addressed by the reviewers.

Reviewer #3 (Remarks to the Author):

JHU-083 is a pro-drug for DON which has demonstrated remarkable potency in treating a variety of tumors in mice. In this work, the authors explore the effect of JHU083 on a murine model of tuberculosis. The authors show that treatment of infected mice has a modest effect on bacterial burdens in organs of infected accompanied by a dramatic effect on survival, body weight and lung weight of infected mice suggesting that something important is happening. The authors analyze different cell types in the lung which point to a role of myeloid derived suppressor cells. In summary, the results are interesting and point to a host-directed benefit of JHU083 in chemotherapy of Mtb-infected mice. The authors have addressed the major concerns and generated a more focused demonstration of JHU083 efficacy and possible mechanism of action. It is a pity that the authors did not demonstrate DON release from JHU083 by Mtb but this seems minor compared to the overall important demonstration of JHU083 efficacy.

POINT BY POINT RESPONSE TO THE REVIEWERS

NOTE TO THE REVIEWERS (SECOND REVISION)

We appreciate the reviewers for going through the manuscript once again and approving the changes. We believe that these suggestions have significantly contributed to improving the quality of the manuscript further. Thank you.

Reviewer #1 (Remarks to the Author):

The concerns I had raised have been adequately addressed by the reviewers.

Thank you so much for the helpful suggestions.

Reviewer #3 (Remarks to the Author):

JHU-083 is a pro-drug for DON which has demonstrated remarkable potency in treating a variety of tumors in mice. In this work, the authors explore the effect of JHU083 on a murine model of tuberculosis. The authors show that treatment of infected mice has a modest effect on bacterial burdens in organs of infected accompanied by a dramatic effect on survival, body weight and lung weight of infected mice suggesting that something important is happening. The authors analyze different cell types in the lung which point to a role of myeloid derived suppressor cells. In summary, the results are interesting and point to a host-directed benefit of JHU083 in chemotherapy of Mtb-infected mice.

The authors have addressed the major concerns and generated a more focused demonstration of JHU083 efficacy and possible mechanism of action. It is a pity that the authors did not demonstrate DON release from JHU083 by Mtb but this seems minor compared to the overall important demonstration of JHU083 efficacy.

Thank you so much for all the helpful suggestions. We agree with the reviewer's point that we did not show the DON release from JHU083 by *Mtb*. Interestingly, unlike other JHU083-related prodrugs that are in clinical trials, JHU083 prodrug undergoes spontaneous and rapid conversion to DON at 37°C (*Slusher, personal email communication*). This spontaneous conversion is expected to substantially interfere with measurement of *Mtb*-mediated conversion of JHU083 to DON. We plan to study *Mtb*-mediated conversion whenever an improved JHU083-related prodrug becomes available for preclinical testing.